# Structural basis of the interaction between BCL9-Pygo and LDB-SSBP complexes in assembling the Wnt enhanceosome

Hongyang Wang [1], Mariann Bienz [2], Xiao-Xue Yan [3] ✉ & Wenqing Xu [1] ✉

The Wnt enhanceosome is responsible for transactivation of Wnt-responsive genes and a promising therapeutic target for treatment of numerous cancers with Adenomatous Polyposis Coli (APC) or β-catenin mutations. How the Wnt enhanceosome is assembled remains poorly understood. Here we show that B-cell lymphoma 9 protein (BCL9), Pygopus (Pygo), LIM domain-binding protein 1 (LDB1) and single-stranded DNA-binding protein (SSBP) form a stable core complex within the Wnt enhanceosome. Their mutual interactions rely on a highly conserved N-terminal asparagine proline phenylalanine (NPF) motif of Pygo, through which the BCL9-Pygo complex binds to the LDB-SSBP core complex. Our crystal structure of a ternary complex comprising the N-terminus of human Pygo2, LDB1 and SSBP2 reveals a single LDB1-SSBP2 complex binding simultaneously to two Pygo2 molecules via their NPF motifs. These interactions critically depend on the NPF motifs which bind to a deep groove formed between LDB1 and SSBP2, potentially constituting a binding site for drugs blocking Wnt/β-catenin signaling. Analysis of human cell lines lacking LDB or Pygo supports the functional relevance of the Pygo-LDB1-SSBP2 interaction for Wnt/β-catenin-dependent transcription.

The Wnt/β-catenin signaling pathway is highly conserved from the most primitive animals to humans and controls cell fate decisions during embryonic development[1] and in adult stem cell compartments[2]. Mutations of Wnt pathway components, including β-catenin and APC, lead to constitutive transcriptional activation of Wnt target genes and are tightly associated with many cancers, including the vast majority of colorectal cancers. The β-catenin-dependent Wnt transcriptional activation complex, also known as the Wnt enhanceosome, is probably the most promising drug target for therapeutic intervention in cancers driven by inactivating APC or activating β-catenin mutations[3,4]. In this multiprotein transactivation complex, the N-terminal region of β-catenin binds to BCL9 and Pygopus (Pygo), nuclear factors that were shown to be important for β-catenin-dependent transcription of Wnt target genes[5–7]. We have previously revealed how β-catenin binds to BCL9 and how BCL9 binds to Pygo[8,9] (Supplementary Fig. 1). However,

it remains unclear how Pygo interacts with the Wnt enhanceosome as a whole.

Mammals possess two Pygo proteins, Pygo1 and Pygo2[7], whereby in the mouse, Pygo2 is expressed much more widely than Pygo1[10]. *Pygo1* homozygous null mice are viable and fertile, whereas *Pygo2* homozygous null mice die shortly after birth with defects in morphogenesis of brain, eyes, hair follicles, kidney and lung[11–13]. The defects during lung and mammary gland development in *Pygo2* null mice are reminiscent of the effects observed in mice overexpressing the Wnt inhibitor Dickkopf-1[12,14,15], consistent with a role of Pygo2 as an important transcriptional co-activator of Wnt target genes. This was consolidated by conditional knock-outs of Pygo1/2 and Bcl9/B9l in the murine intestinal epithelium, which bias cell fates in normal crypts from proliferative to differentiated and synergize in *Apc*-mutant tumors to reverse their transcriptional program from stem cell-like

[1]School of Life Science and Technology, ShanghaiTech University, Shanghai, China. [2]Medical Research Council Laboratory of Molecular Biology, CB2 0QH Cambridge, United Kingdom. [3]National Laboratory of Biomacromolecules, Chinese Academy of Sciences Center for Excellence in Biomacromolecules, Institute of Biophysics, Chinese Academy of Sciences, Beijing, China. ✉e-mail: snow@ibp.ac.cn; xuwq2@shanghaitech.edu.cn

towards normal[16]. Pygo2 is overexpressed in various cancers, including colorectal, breast, lung and prostate cancer, and its overexpression is linked to poor prognosis[17–20]. Inhibition of Pygo2 expression in cancer cells suppresses their proliferation and invasiveness and decreases the expression of Wnt target genes[17,21]. It was therefore suggested that Pygo2 could serve as a potential prognostic biomarker and therapeutic target in these cancers[17–20]. This notion was strongly reinforced by the discovery that the life span of mice bearing *Apc* *Min* mutant tumors was dramatically extended by simultaneous loss of Pygo1/2 and Bcl9/B9l in their intestine and that Bcl9/B9l loss alone restored a normal life in *Apc* *1322T* mutant mice and essentially cured them of their neoplastic disease[16].

Pygo proteins contain three highly conserved domains: a nuclear localization signal (NLS), an N-terminal homology domain (also called N-box) including a highly conserved NPF motif and a C-terminal plant homology domain (PHD) (Fig. 1a)[6,7]. The PHD finger of Pygo binds to histone H3 tail methylated at lysine 4, consistent with its co-activator function, but this domain is also necessary and sufficient to bind to the homology domain 1 (HD1) of BCL9[9] which, in turn, binds to β-catenin through a second homology domain (HD2)[8], thereby functioning as an adaptor between Pygo and β-catenin. In contrast to the structured PHD finger, the N-terminus of Pygo is intrinsically disordered and spans the NLS as well as a short stretch of prolines and the NPF motif, each highly conserved amongst Pygo ortholog[7,22–24]. A single point mutation within this motif (Phe to Ala) abolishes the transcriptional activity of Pygo in cell-based assays and strongly reduces Wnt signaling in vivo[22,23], underscoring the functional importance of this motif.

The binding partner of the NPF motif has recently been discovered to be an ancient protein complex composed of the transcriptional adaptor Chip/LDB (LIM domain-binding protein) and single-stranded DNA binding protein (SSBP2/3/4), also known as ChiLS, which specifically to bind to the NPF motif of Pygo proteins[25]. In various cellular contexts, ChiLS appears to constitute the structural core of the Wnt enhanceosome[25] and it regulates gene transcription by mediating long-range enhancer-promoter interactions[26–28]. Notably, ChiLS not only binds to the NPF motif within the Pygo N-terminus, but also to a similar motif of Osa/ARID1A[25], the DNA-binding subunit of the BAF chromatin remodeling complex and important tumor suppressor[29]. ChiLS adopts a rotationally symmetrical $SSBP_2$-$LDB_2$-$SSBP_2$ architecture[30,31], and may thus bind simultaneously to two different NPF-containing proteins. In addition, ChiLS uses a separate domain called LIM interaction domain (LID) to bind to various tissue-specific DNA-binding proteins[25]. ChiLS can thus integrate these signals from multiple transcription factors and translate them into ON or OFF states of the target genes[25,31,32]. The molecular basis for the recognition of NPF motif-containing proteins by ChiLS remains unknown.

Here, we show that BCL9, Pygo, LDB1 and SSBP form a stable subcomplex in the Wnt enhanceosome and illustrate how Pygo binds to the ChiLS core complex. Our work reveals how the critical NPF motif of Pygo tethers it to ChiLS to regulate β-catenin-dependent transcription of Wnt target genes. We also highlight that the NPF-binding groove may provide a promising target for future discovery of therapeutic drugs in cancer.

## Results

### BCL9, Pygo, LDB1 and SSBP form a stable complex
To reveal how the Wnt enhanceosome forms, we tested overexpression and purification of various recombinant full-length protein components of the human Wnt enhanceosome (Fig. 1a). BCL9, Pygo2 and LDB1 individually are poorly behaved. In contrast, co-expression of full-length BCL9 and Pygo2, or full-length LDB1 and SSBP2, respectively, allowed us to purify both binary complexes from HEK293S GnTI⁻ cells (Supplementary Fig. 2). Both BCL9-Pygo2 and LDB1-SSBP2 (ChiLS, for short) complexes appeared to be stable in solution and, importantly, can directly interact with each other and co-migrate in size-

exclusion chromatography (Fig. 1b). Using a Biolayer Interferometry (BLI) assay, we measured the binding affinity between the recombinant BCL9-Pygo and ChiLS complexes to be 32 nM (Fig. 1c, d). Thus, we conclude that BCL9, Pygo2, LDB1 and SSBP2 proteins can form a stable subcomplex within the Wnt enhanceosome.

It has been reported that the BCL9 HD3 domain can engage in a weak interaction with the ChiLS complex[32]. To examine the role of BCL9 HD3 domain in the BCL9-Pygo-LDB1-SSBP2 complex, we purified recombinant BCL9 proteins bearing HD1 and HD3 deletions, respectively. While deletion of HD1 completely disrupted complex formation, HD3-deleted BCL9 can fully support assembly of the BCL9-Pygo-LDB1-SSBP2 complex (Fig. 1c, d). Thus, the BCL9-Pygo-LDB1-SSBP2 complex formation is predominantly mediated by the interactions between BCL9-HD1 and Pygo on the one hand, and between the Pygo N-terminus and ChiLS on the other hand (Fig. 1e).

### Biochemical characterization of the Pygo2-ChiLS interaction
As mentioned above, the Pygo N-terminus contains an NLS and two high conserved elements separated by a short linker sequence: a proline cluster (PPP, amino acids 63–65) and an NPF motif (amino acids 76–78) (Fig. 2a). To test whether both elements contribute to the binding between Pygo2 and ChiLS, we overexpressed and purified several GST-tagged N-terminal fragments from human Pygo2. GST pull-down assays with purified ChiLS complex containing LDB1(56–285) and SSBP2(1–94) revealed that this complex interacts efficiently with Pygo2 fragments containing both elements including Pygo2(58–84), and also with *Drosophila* Pygopus(71–107), but only weakly with a shorter Pygo2 fragment lacking PPP, i.e. Pygo2(66–81) (Fig. 2b), indicating a critical role of the proline cluster in this interaction. We also found that Pygo2(58–84) neither binds to a shorter ChiLS complex containing LDB1(56–285) and SSBP2(1–77) nor to LDB1 alone (Supplementary Fig. 3), confirming that the integrity of ChiLS[30], and in particular the interface between SSBP2(78–94) and LDB1[30], are critical for the binding of this complex to an N-terminal Pygo2 fragment spanning the PPP cluster and the NPF motif.

Next, we measured the affinity between Pygo2(58–84) and ChiLS using the BLI assay. This revealed a tight sub-micromolar interaction between Pygo2(58–84) and ChiLS (Kd of ~0.45 μM), but this binding affinity was reduced >10x for the shorter Pygo2 fragment without PPP, i.e. Pygo2(66–81) (Kd of ~11 μM; Fig. 2c), confirming the importance of the proline cluster. We also note that the conserved NLS upstream of Pygo2(58–84) does not contribute significantly to the Pygo2-ChiLS interaction (Fig. 2b, c).

### Crystal structure of the ternary Pygo2-ChiLS complex
To provide a structural basis for the Pygo2-ChiLS interaction, we determined the crystal structure of the ternary complex assembled from human Pygo2(58–84), LDB1(56–285) and SSBP2(1–94) at 2.45 Å resolution (Table 1). In our crystal lattice, there is one Pygo2 molecule and one LDB1-$(SSBP2)_2$ complex in each asymmetric unit (Fig. 3a). Residues 58–65 and 71–81 of Pygo2(58–84) are visible in the electron density map, and residues 66–70 are in their most probable configuration (Supplementary Fig. 4). Pygo2(58–84) adopts an unstructured loop, confirming secondary structure predictions (Supplementary Fig. 5a). Its N-terminus is located on the surface of the SSBP2 dimer whereas its C-terminus protrudes into a pocket formed by the interface between LDB1 and SSBP2. Taking into account the crystallographic packing patterns and the ability of LDB1 to form homodimers[30,31], we conclude that one ChiLS complex can bind simultaneously to two Pygo molecules (Fig. 3b).

The structure of the ChiLS core complex is almost identical to the previously reported structure of ChiLS[30] (RMSD = 0.366 Å) (Supplementary Fig. 5b), indicating that the binding of Pygo2 to ChiLS does not alter its overall conformation. However, there was one marked conformational difference for a loop of SSBP2 (amino acids 49–51):

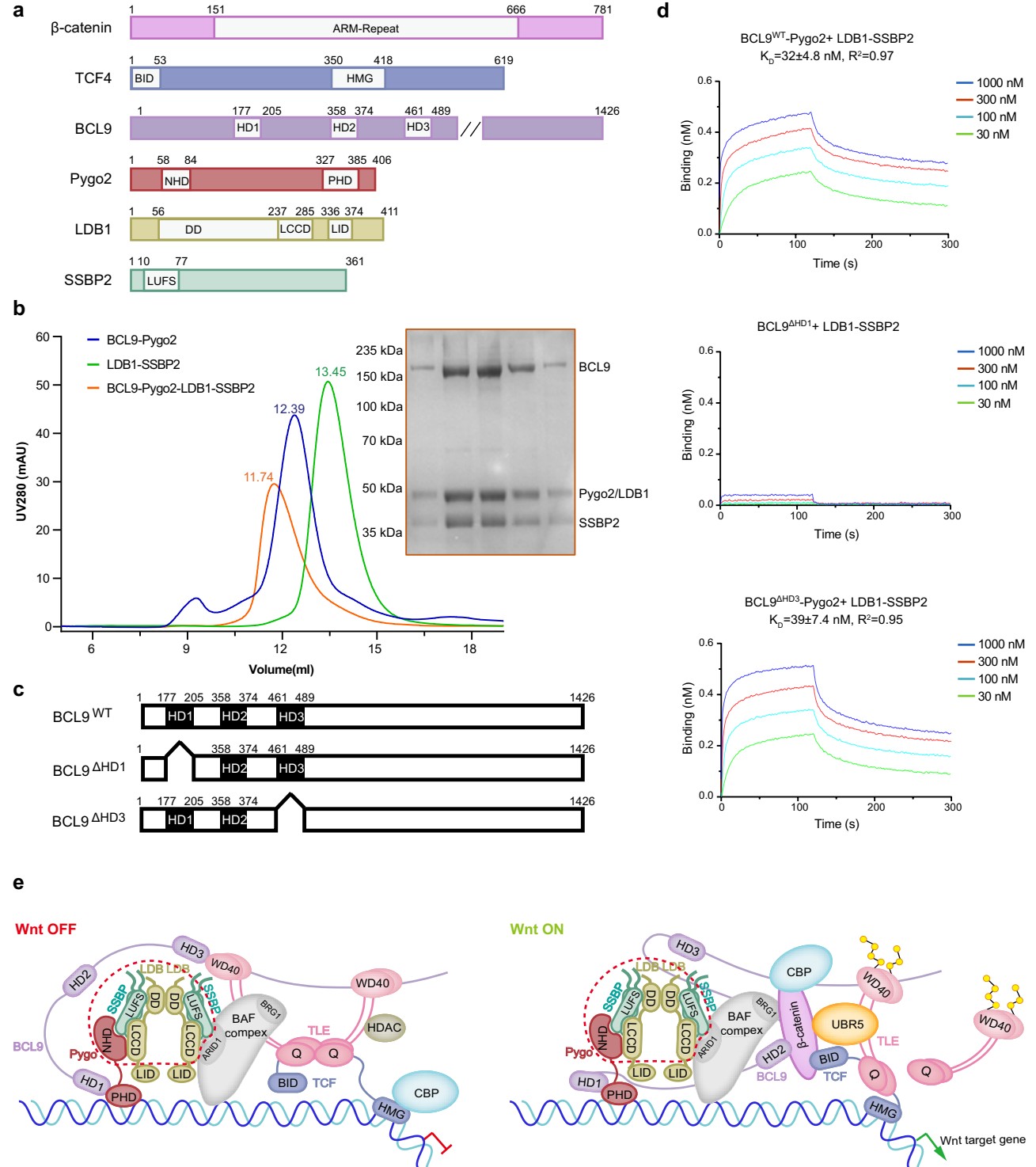

**Fig. 1 | BCL9, Pygo2, LDB1 and SSBP2 form a stable complex in vitro. a** Domain structures of key components of the human Wnt enhanceosome; the double slash indicates shortening of the extensive C-terminus typical of BCL9-related proteins, which contains the binding site for Groucho/TLE and is required for Wnt responses in flies and human cells[25,32]; ARM-repeat (β-catenin Armadillo repeat), BID (β-catenin-binding domain), HMG (high mobility group), HD1/2/3 (homology domain 1/2/3), NHD (N-terminal homology domain of Pygo comprising an NLS, the PPP cluster and NPF motif), PHD (plant homology domain), DD (dimerization domain), LCCD (LDB/Chip conserved domain), LID (LIM-interaction domain), LUFS (LUG/LUH, Flo8 and SSBP conserved domain). **b** Co-migration of BCL9-Pygo2 and LDB1-SSBP2 complexes (both consisting of full-length human proteins) on size-exclusion

chromatography (using a Superose 6 increase 10/300 GL column). Experiments were independently performed three times with similar results. **c** Diagram of human wt or truncated BCL9 tested for interaction with LDB1-SSBP2. **d** Binding-affinities of full-length or truncated forms of Avi-tagged BCL9-Pygo2 with full-length LDB1-SSBP2 complex, as measured by BLI assays. The $K_D$ was calculated based on steady-state analysis and was presented as mean values ± standard deviations (SD). $n = 4$ of analyzed concentrations. **e** A schematic model of the Wnt enhanceosome in its OFF or ON state (see also refs. 32,52); the red dashed circle marks the Pygo2-LDB1-SSBP2 ternary complex whose crystal structure is reported here. Note also that the HMG domain of TCF bends DNA[53,54], which may facilitate assembly of transcriptional complex. Source data are provided as a Source Data file.

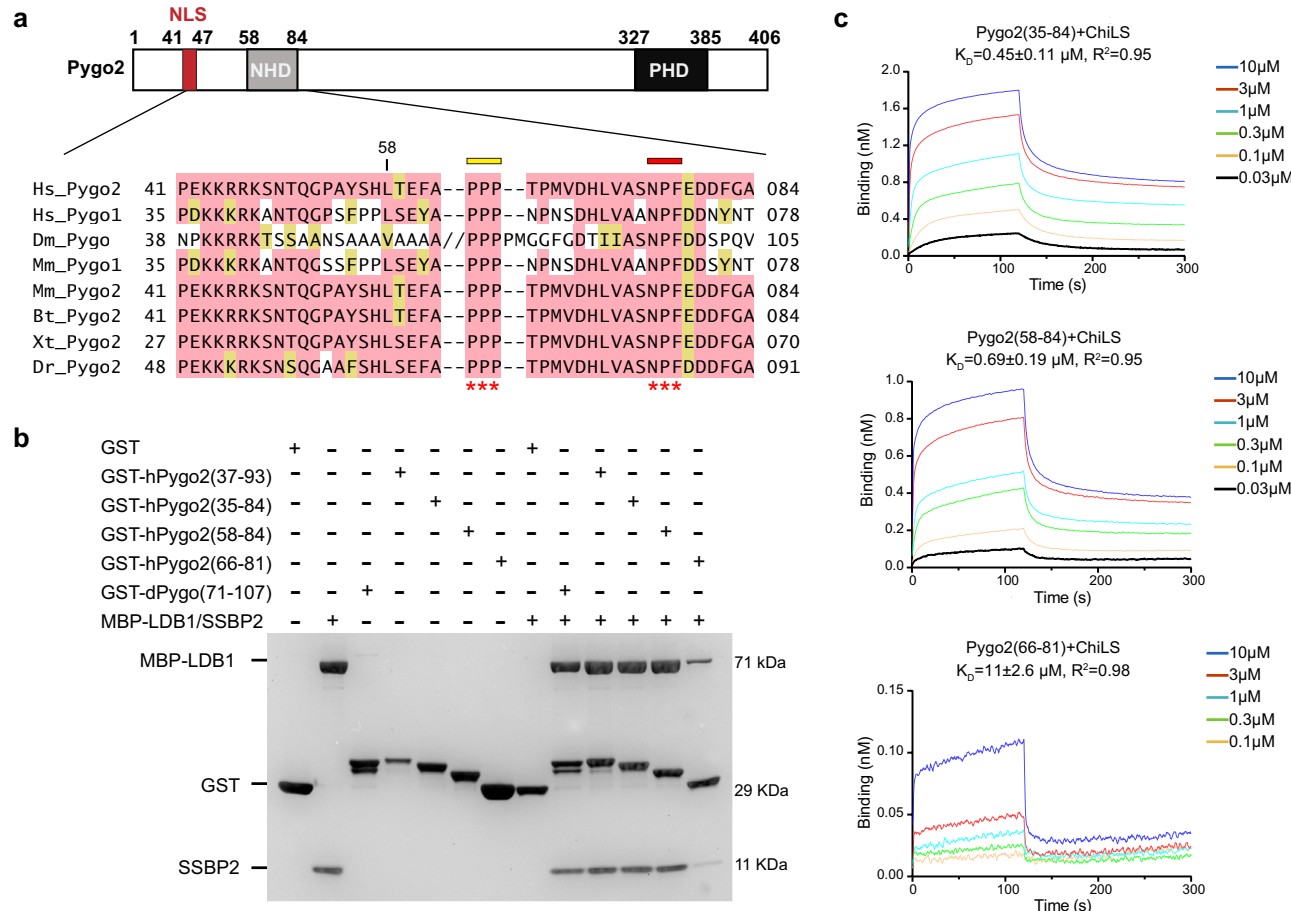

**Fig. 2 | Biochemical characterization of the interaction between Pygo2 and ChiLS. a** A schematic diagram of the conserved domains of human Pygo2 and sequence alignment of the Pygo N-terminus, in which two conserved elements (PPP cluster and NPF motif) are indicated with asterisks. The double slash in the Dm_Pygo sequence represents a 22-residue insertion. Abbreviations: Hs *Homo sapiens*, Dm *Drosophila melanogaster*, Mm *Mus musculus*, Bt *Bos taurus*, Xt *Xenopus tropicalis*, Dr *Danio rerio*. **b** Functional tests of the Pygo-ChiLS interaction, as shown by GST pull-down assays. The SDS-PAGE gel was stained with Coomassie Blue.

Experiments were independently performed three times with similar results. Labeled are the molecular weights (MW) of the corresponding proteins. **c** Binding affinities of three Pygo2 fragments for ChiLS, as measured by BLI assays. The $K_D$ was calculated based on steady-state analysis and was presented as mean values ± standard deviations (SD). $n = 6$ of analyzed concentrations, except for Pygo2(66–81) ($n = 5$ of analyzed concentrations). Source data are provided as a Source Data file.

a lysine residue within this loop (K50) shifted by about 6.6 Å, tilting away from the bound Pygo2 (Supplementary Fig. 5c), thereby avoiding a clash with Pygo2 M68 and L72.

## The interface between Pygo2 and ChiLS
Pygo2(58–84) interacts with ChiLS through three sets of contacts (Fig. 4). Firstly, F68, W69 and Y72 of one SSBP2 molecule and Y25, F60 and W64 of another, together with L245 and I248 of LDB1, form a deep hydrophobic pocket that interacts with NPF (residues 76–78) of Pygo2(58–84) (Fig. 4a). Notably, SSBP2 F60 corresponds to SSDP F58 which was previously proposed to play a key role in defining the Pygo-binding pocket of ChiLS[31]. Secondly, the PPP cluster (residues 63–65) and L72 of Pygo2(58–84) are accommodated by a hydrophobic groove formed largely by W48 and H62 of SSBP2 (Fig. 4b). Thirdly, LDB1 R232 and R244 undergo tight hydrogen bonds with D80 within the NPF motif of Pygo2, and we also note salt bridges between Pygo2 E79 and LDB1 N237 and N241 (Fig. 4c). In total, the Pygo2-ChiLS interface buries an area of 1273.1 Å², a rather extensive interface which explains the relatively high binding affinity between Pygo2(58–84) and ChiLS. The human Pygo1 paralog and *Drosophila* Pygopus are likely to interact with ChiLS in a similar manner, since the Pygo2 residues involved in the ChiLS interaction are highly conserved in these Pygo relatives (Fig. 2a).

## Identification of hotspots in the Pygo2-ChiLS interface
To assess the significance of individual interactions observed in our structure and identify interface hotspots for potential drug development, we tested various Pygo2 and ChiLS missense mutants in pull-down and BLI assays. First, we generated 13 mutants in Pygo2 residues contacting the ChiLS interface that reside in PPP or NPF (Fig. 5, Supplementary Fig. 6). A triple PPP > AAA mutation strongly reduces the binding of Pygo2 to ChiLS, whereas single or double point mutations of the PPP cluster have a lesser effect (Fig. 5a, c; Supplementary Fig. 7). This is consistent with our previous data that this cluster increases the binding affinity between Pygo and ChiLS (Fig. 2). However, a single alanine substitution of the N-terminal residue of NPF (N76A) strongly reduces the binding of Pygo2 to ChiLS, while the single P77A and F78A mutations completely abolish this interaction in both GST pull-down and BLI assays (Fig. 5a, c; Supplementary Fig. 7), consistent with previous work[25]. This underscores the central importance of these two near-invariant residues of the NPF motif and is also consistent with our crystal structure which shows that P77 and F78 are closer to the hydrophobic core of ChiLS than N76 (Fig. 4a). By contrast, mutation of D80A (engaging in multiple hydrogen bonds with LDB1; Fig. 4c) only slightly decreases the ability of Pygo2 to interact with ChiLS (Fig. 5a, c; Supplementary Fig. 7) while two further mutations in down-stream adjacent acidic residues (E79A, D81A) do not affect binding, consistent

**Table 1 | Data collection and refinement statistics**

| | Pygo2-LDB1-SSBP2 |
|---|---|
| Data collection | |
| Space group | $P6_522$ |
| Cell dimensions | |
| $a, b, c$ (Å) | 104.5, 104.5, 250.3 |
| $\alpha, \beta, \gamma$ (°) | 90, 90, 120 |
| Resolution (Å) | 20.0–2.45 (2.52–2.45)[a] |
| $R_{merge}$ | 6.3 (191.1) |
| $I/\sigma I$ | 44.86 (1.75) |
| Completeness (%) | 100.0 (100.0) |
| Redundancy | 12.7 (13.4) |
| Refinement | |
| Resolution (Å) | 19.93–2.45 |
| No. reflections | 29241 |
| $R_{work}/R_{free}$ | 21.38/25.69 |
| No. atoms | |
| Protein | 3241 |
| Water | 67 |
| $B$-factors | |
| Protein | 50.373 |
| Water | 71.700 |
| R.m.s. deviations | |
| Bond lengths (Å) | 0.005 |
| Bond angles (°) | 1.374 |

[a]Values in parentheses are for highest-resolution shell. The diffraction data for this structure is from 1 crystal.

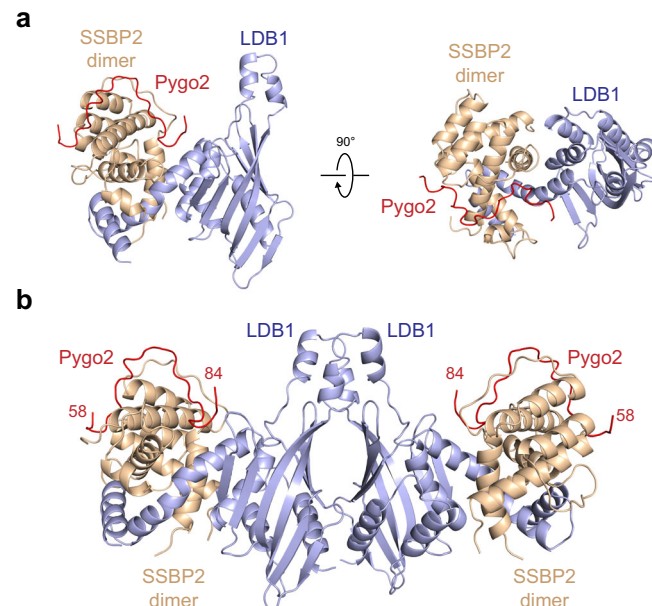

**Fig. 3 | Crystal structure of the Pygo2-LDB1-SSBP2 complex. a** Two orthogonal views of the complex assembled from human component proteins; red, Pygo2; light blue, LDB1; wheat, SSBP2. **b** Structure of a single ChiLS core complex, simultaneously bound to two Pygo2 molecules.

with the low degree of sequence conservation in these residues. Finally, we also found that Pygo2 L72A (a highly conserved residue between PPP and NPF; Fig. 2a) also strongly reduces the binding between Pygo2 and ChiLS (Fig. 5a), but we did not observe any significant effects on binding for Pygo2 F61A, T66A and P67A (Supplementary Fig. 6). These results are highly consistent with previous results derived from cell-based assays that uncovered key residues of *Drosophila* Pygopus for its co-activator function[22], implicating the interface between Pygo and ChiLS in transcriptional activation.

We next tested 7 SSBP2 mutants and 5 LDB1 mutants tagged with maltose-binding protein (MBP) for their binding to Pygo2. MBP pull-down and BLI assays revealed that any mutations of SSBP2 residues that are in direct contact with Pygo almost completely blocked the ChiLS-Pygo2 interaction (except for Pygo2 Y72A; Fig. 5b, c; Supplementary Fig. 7), consistent with previous Co-IP assays in LDB1/2 double knock-out (DKO) HEK293T cells[31]. Similarly, an alanine substitution of LDB1 (R244A) blocks the ability of ChiLS to interact with Pygo2, in contrast to N237A, N241A, L245A and I248A which do not (Fig. 5b, c; Supplementary Fig. 7). This is consistent with our crystal structure which revealed a close (2.8 Å) hydrogen bridge between LDB1 R244 and Pygo2 D80 (Fig. 4c), ascribing a key role to this arginine residue of LDB1 in the ChiLS-Pygo interaction.

### The Pygo2-ChiLS interaction is important for Wnt responses in human cells

To test the role of the Pygo2-ChiLS interaction in β-catenin-dependent transcription, we generated DKO HEK293T cells lacking LDB1/2 (LDB⁻/⁻) or Pygo1/2 (Pygo⁻/⁻) by CRISPR engineering[33] (Supplementary Fig. 8). Based on our mutational analysis (Fig. 5), we chose the Pygo2 triple mutant NPF > AAA and the LDB1 point mutant R244A whose ability to form a ternary Pygo2-ChiLS complex is abolished, to monitor their activity in stimulating Wnt/β-catenin target genes in complementation assays of DKO cells. We employed the TCF-dependent TOPflash luciferase reporter assay[34] to measure the transcriptional activity of wild type (wt), LDB⁻/⁻ and Pygo⁻/⁻ HEK293T cells after Wnt pathway stimulation via the potent GSK3β inhibitor lithium chloride (LiCl)[35].

The LiCl-induced transcriptional activity of these DKO cells is reduced to ~60–90% of wt cells in the case of LDB⁻/⁻ cells and even further in the case of Pygo⁻/⁻ cells (to ~30–50% of wt) (Fig. 6a), as expected from the stimulatory role of these factors in the transcription of Wnt target genes. Transient expression of wt Pygo2-Flag in Pygo⁻/⁻ cells partially restored Wnt-responsiveness whereas the triple NPF > AAA mutant failed to do so (Fig. 6b), consistent with previous results based on the same mutant of *Drosophila* Pygopus[22]. Similarly, expression of wt LDB1-Flag in LDB⁻/⁻ cells restored full Wnt-responsiveness, whereas the R244A mutant proved inactive (Fig. 6c). These results indicate the functional significance of the interaction between Pygo2 and ChiLS in transactivating Wnt target genes.

To identify Wnt target genes regulated by LDB1/2 or Pygo1/2, we performed differentially expressed gene (DEG) analysis via RNA sequencing (RNA-seq) in wt or DKO HEK293T cells with or without LiCl stimulation and validated the RNA-seq results by performing RT-qPCR for 8 randomly selected genes (Supplementary Table 1 and 2; Supplementary Fig. 9). We found that the expression levels of 46 Wnt target genes (most of which are likely to be indirect, as discussed below), out of 577 tested, were significantly altered in these samples (Supplementary Data 1, Supplementary Fig. 10), whereby the expression profile of Pygo⁻/⁻ cells differed markedly from wt cells, more so than that of LDB⁻/⁻ cells (Supplementary Fig. 10). These observations indicate the functional relevance of the Pygo and LDB enhanceosome components for the transcription of a subset of Wnt target genes in this human epithelial cell line.

### Discussion

Genes encoding Wnt pathway core components, in particular APC and β-catenin, are among the most frequently mutated genes in human cancers, especially in colorectal cancer where ~80% of cases are

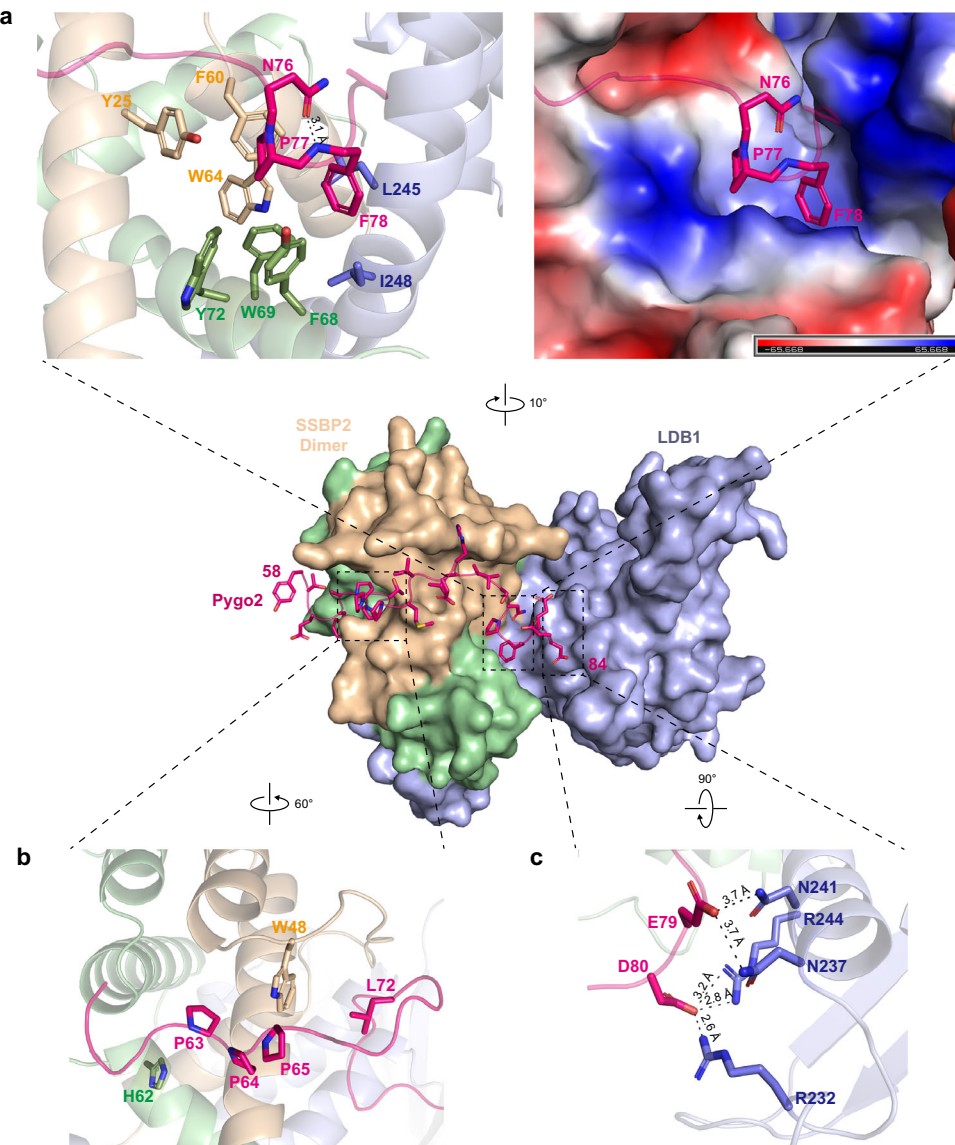

**Fig. 4 | Specific interactions between Pygo2 and ChiLS.** Coloring as in Fig. 3, except for green, marking second SSBP2 molecule. **a** Hydrophobic interactions formed between Pygo2 NPF and its cognate surface of ChiLS; left, ribbon representation; right, electrostatic surface representation. **b** Interface residues mediating interactions between Pygo2 PPP and SSBP2. **c** Hydrogen bonds and salt bridges between the negatively charged side chains of conserved glutamic acid (E) or aspartic acid (D) residues 3′ adjacent to Pygo2 NPF (see also Fig. 2a) and the guanidium or carboxamide groups in the sidechains of conserved arginine (R) or asparagine (N) residues of LDB1.

initiated by Wnt pathway hyperactivation as a result of loss-of-function of the APC tumor suppressor, a negative regulator of the pathway. While a large body of evidence has supported critical roles of abnormal Wnt/β-catenin activation in cancer development and resistance to cancer therapy, there is still no FDA-approved Wnt pathway inhibitor for therapeutic treatment of these cancers. It is widely accepted that the Wnt enhanceosome, the nuclear endpoint of the Wnt signaling cascade down-stream of the β-catenin destruction complex, is the most promising target for treatment of cancers with APC or β-catenin mutations. Here we focus on important Wnt enhanceosome players associated with β-catenin via the critical β-catenin-BCL9 interaction. We show that the binary BCL9-Pygo and LDB1-SSBP2 complexes can directly interact with each other and reveal the structural basis of the interaction between the Pygo N-terminus and its cognate surface in the LDB1-SSBP2 core complex.

In our crystal structure of the Pygo2-LDB1-SSBP2 complex, a portion of the Pygo2 N-terminus is located on the surface of SSBP2 dimer, predicting possible formation of a BCL9-Pygo-LDB1-SSBP2 complex in a 2:2:2:4 molar ratio. Examination of the structural features of the BCL9-Pygo-LDB1-SSBP2 complex by negative-staining EM and cryo-EM methods suggests that this complex per se may not adopt a tightly-packed shape beyond individually folded domains and the subcomplexes formed by them. Instead, its components may serve as flexible recruiters of other transcriptional co-activators and co-repressors, including the NPF-containing ARID1 subunit of the BAF chromatin remodeling complex, the CBP/p300 acetyltransferase[25], the mediator complex[36] and the Groucho/TLE co-repressor and its associated histone deacetylase (Fig. 1e)[25]. All of these factors are likely to contribute to the function of the Wnt enhanceosome in conferring ON and OFF states on Wnt target genes[25,36], whereby the precise composition of this enhanceosome depends not only on Wnt stimulation, but also varies between cell types and over time. The intrinsic symmetry of the LDB1-SSBP2 core complex and its 2:4 stoichiometry allowing for simultaneous binding of two distinct NPF-containing proteins and

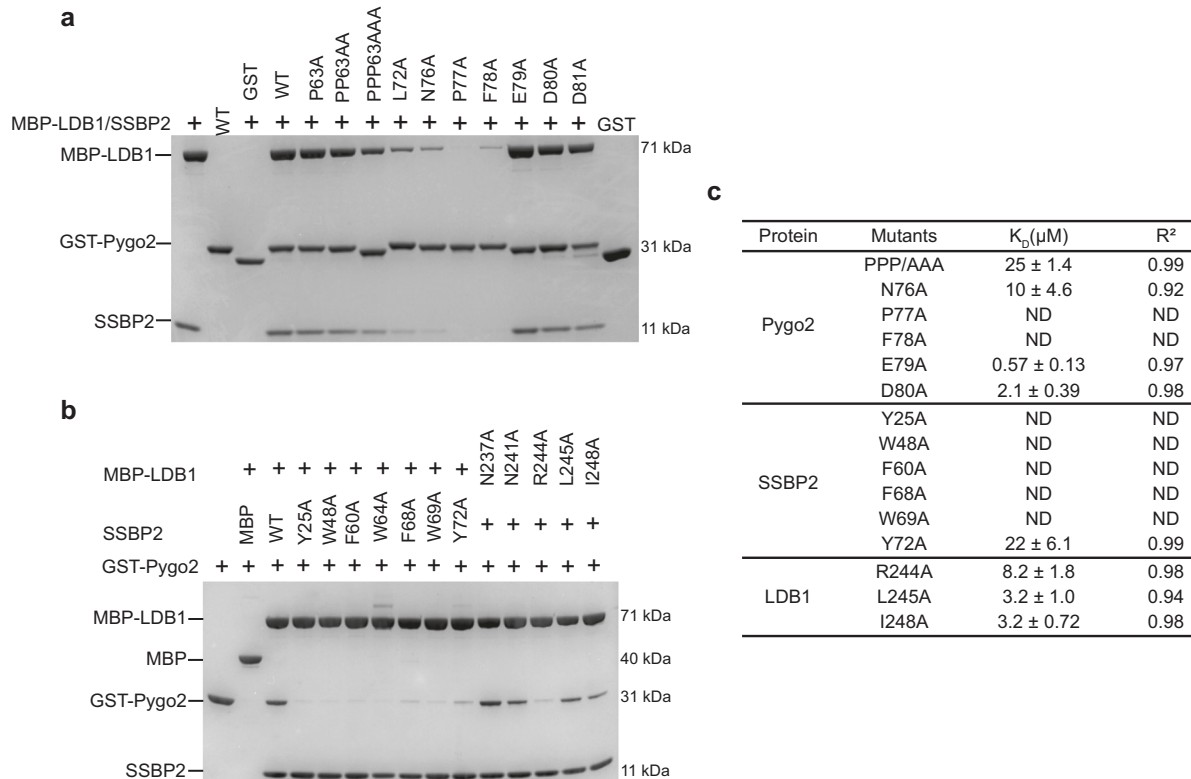

**Fig. 5 | Mutagenesis analysis of the Pygo2-ChiLS interface. a** MBP pull-down assays between MBP-tagged LDB1, SSBP2 and GST-tagged wt or mutant Pygo2 bearing alanine substitutions of individual interface residues; GST, negative control. Experiments were independently performed three times with similar results. Labeled are the molecular weights (MW) of the corresponding proteins. **b** MBP pull-down assays as in **a**, but with mutant MBP-tagged LDB1 or SSBP2 bearing alanine substitutions of interface residues. Experiments were independently performed three times with similar results. Labeled are the molecular weights (MW) of the corresponding proteins. **c** Binding affinities between mutant Pygo2 and wt ChiLS, and vice versa, as measured by BLI assays. ND: no detectable binding. The $K_D$ was calculated based on steady-state analysis and was presented as mean values ± standard deviations (SD). $n = 5$ of analyzed concentrations, except for Pygo2 P77A and SSBP2 F60A ($n = 4$ of analyzed concentrations). Source data are provided as a Source Data file.

| Protein | Mutants | $K_D$(µM) | R² |
|---|---|---|---|
| Pygo2 | PPP/AAA | 25 ± 1.4 | 0.99 |
| | N76A | 10 ± 4.6 | 0.92 |
| | P77A | ND | ND |
| | F78A | ND | ND |
| | E79A | 0.57 ± 0.13 | 0.97 |
| | D80A | 2.1 ± 0.39 | 0.98 |
| SSBP2 | Y25A | ND | ND |
| | W48A | ND | ND |
| | F60A | ND | ND |
| | F68A | ND | ND |
| | W69A | ND | ND |
| | Y72A | 22 ± 6.1 | 0.99 |
| LDB1 | R244A | 8.2 ± 1.8 | 0.98 |
| | L245A | 3.2 ± 1.0 | 0.94 |
| | I248A | 3.2 ± 0.72 | 0.98 |

LID-binding factors is well suited to the fundamental role of the Wnt enhanceosome in integrating multiple inputs from signaling and cell lineages during development and differentiation.

The NPF motif in the Pygo N-terminus is crucial for the transduction of Wnt signaling in vivo[22,23]. A single F > A mutation in this motif in *Drosophila* Pygopus can severely impair its transcriptional activity and lead to death[22]. Similarly, the same mutant can severely reduce the transcription of Wnt target genes such as *sensless*[23]. These studies are highly consistent with our crystal structure of the Pygo2-LDB1-SSBP2 ternary complex, in which the critical NPF motif, in particular the conserved phenylalanine residue, within the Pygo2 N-terminus interacts with a deep hydrophobic groove formed by the LDB-SSBP interface. Our in vitro biochemical analysis using purified N-terminal fragments of Pygo2 also identify individual residues of the NPF motif that are essential for the Pygo-ChiLS interaction. Given the functional significance of NPF (Fig. 5)[22,23], the groove accommodating this motif is an attractive target for developing anti-cancer therapeutics.

The ChiLS complex has been reported to mediate long-range enhancer-promoter interaction, thereby regulating the transcription of master regulatory genes with key functions in embryonic development[25], stem cell maintenance and differentiation along adult cell lineages, including erythroid maturation[37,38]. Our functional assays based on DKO HEK293T cells lacking Pygo1/2 or LDB1/2 (Fig. 6) show that mutations that disrupt the Pygo-ChiLS interaction cannot fully rescue the Wnt response of these null mutant cells. We note that, while the loss of Pygo or LDB is expected to reduce the expression of direct Wnt target genes, these conditions can result in an increased expression in the case of indirect Wnt target genes downstream of Wnt-induced transcriptional repressors (e.g. TLE4; Supplementary Fig. 9), which could explain why many of the genes monitored in our RNA-seq experiments are upregulated as a consequence of Pygo or LDB loss (Supplementary Fig. 10). Furthermore, as mentioned above, the effects of Pygo or LDB loss on Wnt target genes can be complicated by the fact that a single ChiLS core complex can bind simultaneously to Pygo through one arm and to another NPF-containing protein such as ARID1 through the other: recall that ARID1 is the DNA-binding subunit of the chromatin remodeling BAF complex which switches target genes between ON and OFF states, behaving as a repressor of Wnt responses in flies and as an important tumor suppressor in humans[25,29].

Loss of LDB appears to have a smaller effect than loss of Pygo on the transcriptional activity of Wnt target genes in HEK293T cells, which was somewhat unexpected, given the stoichiometry of these factors in the Wnt enhanceosome. We believe that the most likely explanation lies in our use of the TOPflash reporter which, although convenient and readily quantifiable, is not ideal for assaying the transcriptional activity of the Wnt enhancesome, given its key function in mediating the communication between distal enhancers and core promoter elements[25]. This is particularly true for the LDB scaffold protein whose fundamental function in mediating enhancer-promoter communication over long distances (i.e. tens of kilobases) was discovered in the context of cell type specification[39,40], long before this factor was implicated in Wnt responses. However, the TCF binding sites in the TOPflash reporter are directly upstream of its short minimal promoter, which likely explains the modest reliance of this reporter on the long-distance bridging function of LDB. It is also worth bearing in mind that, while Pygo knockout (or knockdown) in different cancer cell lines

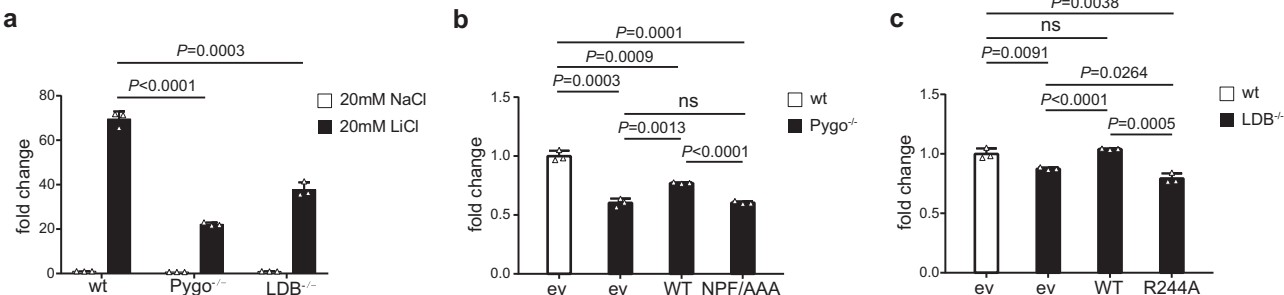

**Fig. 6 | The Pygo-ChiLS interaction is important for transcriptional Wnt responses in human HEK293T cells. a** TOPflash assays in wt or LDB⁻/⁻ or Pygo⁻/⁻ HEK293T cells incubated with 20 mM NaCl or LiCl ±4 h prior to cell lysis; fold changes (Y axis) represent mean values from 3 biological repeats; error bars indicate standard deviations; Unpaired two-tailed t test were used for statistical analyses. **b** TOPflash assays in wt or Pygo⁻/⁻ HEK293T cells transiently transfected with wt or mutant Pygo2 for 24 h and incubated with 20 mM NaCl or LiCl ±6 h prior to cell lysis; fold changes (Y axis) represent mean values from 3 biological repeats; error bars indicate standard deviations; Unpaired two-tailed t test were used for statistical analyses. ev, empty vector control; ns, not statistically significant. **c** TOPflash assays in wt or LDB⁻/⁻ HEK293T cells, transfected and incubated with NaCl or LiCl as in **b**; fold changes (Y axis) represent mean values from 3 biological repeats; error bars indicate standard deviations; Unpaired two-tailed t test were used for statistical analyses. Source data are provided as a Source Data file.

failed to completely abolish the transactivation of TOPflash or selected endogenous Wnt target genes, depletion of Pygo2 protein alone was sufficient to inhibit their growth and to attenuate their oncogenic progression[7,17–19,41–44].

Components of the Pygo-ChiLS complex have been implicated in the neoplastic transformation of multiple cell types in the colon, lung breast and other tissues. As mentioned in the Introduction, while the intestinal-specific knock-out of Pygo1/2 or Bcl9/B9l has little impact on normal intestinal homeostasis, these knock-out conditions suppressed tumorigenesis and extended the disease-free life of *Apc*-mutant mice owing to a normalization of the transcriptional program within their intestinal tumors from stem cell-like to differentiated[16,45,46]. Therefore, cancers driven by Wnt hyperactivity or Pygo2 overexpression may rely substantially on the Pygo-ChiLS-containing core complex of the Wnt enhanceosome for the transcriptional activity of cancer-relevant Wnt target genes[36,47]. Our structure of this complex could guide the development of selective Wnt inhibitors that target Wnt signaling in Pygo-dependent cancers.

## Methods
### Protein cloning, expression and purification
Full-length open reading frames of human BCL9, Pygo2, LDB1 and SSBP2 were sub-cloned into pCAGGS vectors containing a C-terminal Flag tag. The sequences were analyzed by SnapGene 4.2.4. Primers for cloning are shown in Supplementary Table 3. HEK293S GnTI⁻ cells (ATCC) were co-transfected using PEI with plasmids encoding human BCL9 and Pygo2, or with human LDB1 and SSBP2. Upon culture at 37 °C for 72 h, cells were harvested and lysed in lysis buffer containing 50 mM HEPES pH 8.0, 300 mM NaCl, 0.3% CHAPS, 2 mM MgCl₂, 0.25 mM EDTA, 2 mM DTT, 1 mM PMSF supplemented with cOmplete™ Protease Inhibitor Cocktail (Roche) at 4 °C for 30 min. The lysate was clarified by centrifugation at 26,000 g for 30 min at 4 °C. The supernatant was incubated with Flag beads for 2 h, and proteins were eluted with buffer containing 20 mM HEPES pH 8.0, 300 mM NaCl, 2 mM DTT, 0.2 mg/ml flag peptide. Eluted proteins were further purified by size exclusion chromatography using Superose 6 increase 10/300 GL column (GE Healthcare) equilibrated with buffer containing 20 mM HEPES pH 8.0, 200 mM NaCl, 2 mM DTT.

Gene fragments encoding *Drosophila* Pygo(71–107), human Pygo2(37–93), Pygo2(35–84), Pygo2(58–84) and Pygo2(66–81) proteins were optimized with *E.coli* codon usage, generated by overlap extension PCR and subcloned into pGEX-4T-1 vectors including N-terminal Tobacco Etch virus (TEV) protease sites for removal of GST tags, respectively. His-MBP-LDB1(56–285) and His-SSBP2(1–94) or His-SSBP2(1–77) were co-cloned into pETDuet-1 vector including N-terminal

TEV protease sites for removal of His-MBP or His tags. cDNA of human LDB1(56–287) was subcloned into pMAL vectors with an N-terminal His tag. Primers for cloning are shown in Supplementary Table 3. *E.coli* BL21 (DE3) was used for protein overexpression. Briefly, *E.coli* BL21 (DE3) transformed with an expressing plasmid was cultured in Luria broth (LB) at 37 °C to OD₆₀₀ ~ 0.8, and overexpression of recombinant proteins was induced by adding isopropyl β-D-thiogalactoside (IPTG) to a final concentration of 0.2 mM at 18 °C for 16–18 h.

Harvested bacteria were resuspended in lysis buffer (50 mM Tris-HCl pH 8.0, 500 mM NaCl, 2 mM DTT) and homogenized via sonication on ice. Lysates were cleared by centrifugation (26,000 g for 1 h at 4 °C), and the supernatants were incubated with glutathione sepharose 4B resin (GE Healthcare) for 2 h at 4 °C, and the mixtures were then loaded onto an empty column (to collect the resin), washed with lysis buffer, and proteins were eluted with lysis buffer containing 20 mM reduced glutathione. Eluted proteins were further purified by size exclusion chromatography using Superdex 200 increase 10/300 GL column (GE Healthcare) equilibrated with buffer containing 20 mM Tris-HCl pH 8.0, 100 mM NaCl, 2 mM DTT.

For the purification of human Pygo2(58–84) peptides used for crystallization, the N-terminal GST tag was removed on a column by incubating with TEV protease at 4 °C overnight after the GST column was washed with lysis buffer. The flowthrough which contains untagged proteins was collected, diluted five fold with buffer containing 50 mM Tris-HCl pH 8.5, 2 mM DTT and further purified by a HiTrap Q column (GE Healthcare) equilibrated with buffer A (20 mM Tris-HCl pH 8.5, 20 mM NaCl, 2 mM DTT). Proteins were eluted with a 0–100% gradient of buffer B (20 mM Tris-HCl pH 8.5, 1 M NaCl, 2 mM DTT) in 20 column volumes. The protein-containing fractions were collected and concentrated and finally purified by Superdex 75 increase 10/300 GL column (GE Healthcare) using buffer containing 20 mM Tris-HCl pH 8.0, 100 mM NaCl, 2 mM DTT.

Human LDB1(56–285)-SSBP2(1–94) complex for biochemical experiments and crystallization was purified as described[30]. In brief, after sonicated, the cell lysate was cleared by centrifugation (26,000 g for 1 h). The supernatant was loaded onto Ni-NTA affinity resin (Qiagen) and the complex was eluted with lysis buffer containing 300 mM imidazole pH 8.0. After cleavage of His-MBP and His tag with TEV protease at 4 °C overnight, the LDB1-SSBP2 complex was loaded onto a HiTrap Q column (GE Healthcare) equilibrated with buffer A, and was eluted with a 0–100% gradient of buffer B, followed by gel filtration (Superdex 200 10/300 GL, GE Healthcare) equilibrated with buffer containing 20 mM Tris-HCl pH 8.0, 100 mM NaCl, 2 mM DTT for further purification. After concentrated to ~2 mg/mL, the LDB1/SSBP2 complex was flash-frozen in liquid nitrogen, and stored at −80 °C until use.

For purification of His-MBP-LDB1(56–287) protein, after sonication the cell debris was removed by centrifugation for 1 h at 26,000 g. The supernatant was loaded onto Ni-NTA affinity resin (Qiagen) and the LDB1 proteins were eluted with lysis buffer containing 300 mM imidazole pH 8.0. LDB1 was further purified by a Superdex 200 10/300 GL size exclusion column (GE Healthcare) equilibrated with a buffer containing 20 mM Tris-HCl pH 8.0, 100 mM NaCl, 2 mM DTT.

Pygo2, LDB1 or SSBP2 mutants were generated by site-directed PCR mutagenesis and were subcloned, overexpressed and purified in the same way as the wt proteins. Primers for cloning are shown in Supplementary Table 3.

### Assembly of the BCL9-Pygo2-LDB1-SSBP2 complex

BCL9-Pygo2 complex containing full-length human proteins was incubated overnight with LDB1-SSBP2 complex containing full-length human proteins at a 1:2 molar ratio at 4 °C. The mixture was then loaded onto a Superose 6 increase 10/300 GL column (GE Healthcare) to remove excess LDB1-SSBP2 complex. Complex-containing fractions were collected, concentrated, flash-frozen in liquid nitrogen and stored at −80 °C until use.

### Biotinylation of BCL9 proteins

Full-length human BCL9 was sub-cloned into pCAGGS containing a C-terminal Avi tag for biotinylation and a Flag tag for purification. Mutants (hBCL9$^{\triangle HD1}$ (lacking hBCL9(177–205) fragment) and hBCL9$^{\triangle HD3}$ (lacking hBCL9(461–489) fragment) were generated from parental plasmids using standard site-directed mutagenesis and confirmed by sequencing. Primers for cloning are shown in Supplementary Table 3. HEK293S GnTI− cells were co-transfected using PEI with plasmids encoding wt or mutant human BCL9 with C-terminal Avi and Flag tags and plasmids encoding full-length human Pygo2 with a C-terminal Flag tag only. After culturing at 37 °C for 72 h, the overexpressed proteins were purified in the same way as the proteins used for assembling the BCL9-Pygo2 complex.

For biotinylation, purified Avi-tagged wt BCL9-Pygo2, hBCL9$^{\triangle HD1}$-Pygo2 or hBCL9$^{\triangle HD3}$-Pygo2 complexes were incubated with 5 mM ATP, 10 mM magnesium acetate, 50 μM biotin, and 1 μM home-purified BirA at 4 °C for 16 h, respectively. The mixture was then loaded onto a Superose 6 Increase 10/300 GL column (GE Healthcare) to remove excess biotin and BirA proteins. Fractions containing biotinylated proteins were pooled, aliquoted, flash-frozen in liquid nitrogen and stored at −80 °C before use.

### Crystallization, data collection, and structural determination

Numerous methods to obtain crystals of Pygo2-LDB1-SSBP2 triple complex were tried until, eventually, well-diffracting crystals were obtained using the seeding method. The LDB1-SSBP2 complex crystals were grown as microseeds as described[30]. In brief, the LDB1/SSBP2 complex crystals were grown at room temperature by hanging drop vapor diffusion by mix 1 μl of the protein solution and 1 μl of solution containing 100 mM lithium sulfate monohydrate, 100 mM sodium citrate tribasic dihydrate pH 5.6, 1% v/v PEG400, 10 mM DTT. Pygo2(58–84) peptides were incubated for 2 h with LDB1-SSBP2 complex at a 5:1 molar ratio at 4 °C. The mixture was then centrifuged for 10 min at 4 °C to remove pellets. Pre-equilibrated drops were prepared at room temperature by mixing 1 μl of the protein solution with 1 μl of well solution, which contained 100 mM Li₂SO₄, 100 mM sodium citrate tribasic dihydrate pH 5.6, 1% v/v PEG400 and 10 mM DTT. After 12–24 h, microseed stock of the LDB1-SSBP2 crystals were prepared through crushing LDB1-SSBP2 crystals with crystal crusher (Hampton research), and 0.2 μl of microseed solution was added into the pre-equilibrated drops. Crystals were grown for 2–3 days, and harvested crystals were cryoprotected with 10% PEG 400 and 15% ethylene glycol by using a quick-soak flash-freeze method.

A 2.45 Å data set was collected at the BL19U1 beamline in National Center for Protein Science Shanghai (NCPSS) at a wavelength of 0.9792 Å. The data were integrated and scaled using HKL3000[48]. The space group was $P6_522$ with the unit cell dimensions a = 104.5 Å, b = 104.5 Å, c = 250.3 Å. Molecular replacement was carried out with PHENIX 1.19.2-4158[49], using the LDB1-SSBP2 structure (PDB entry 6TYD, [https://doi.org/10.2210/pdb6TYD/pdb]) as the search model. The electron density for Pygo2 peptide was clear except for residues 66–70. One asymmetric unit contained one Pygo2 molecule, one LDB1 molecule and two SSBP2 molecules. Model building and refinement was performed using Coot 0.8.9.2[50] and Refmac5 in CCP4 7.1.000[51]. In the Ramachandran plot, there are 90.34% most favorable and 9.66% allowed. All structural model figures were generated by PyMOL 2.5.1 (Schrödinger, Inc.). The data collection and refinement statistics are summarized in Table 1.

### In vitro pull-down assays

For GST pull-down assays, 5 μM of wt GST-Pygo fragments (or GST-Pygo mutants), 20 μM of wt MBP-LDB1-SSBP2 complex and 20 μl glutathione sepharose 4B resin (GE Healthcare) were mixed in 100 μl pull-down buffer containing 50 mM Tris-HCl pH 8.0, 100 mM NaCl, and 10 mM DTT. Samples mixtures were incubated at room temperature for 2 h, followed by washing the resin three times with pull-down buffer. During each wash, 200 μl of pull-down buffer was added to each sample and incubated at room temperature for 5 min before centrifugation and removal of supernatant. After washing, the resin was boiled and analyzed by SDS-PAGE with Coomassie blue staining. For MBP pull-down assays, 5 μM of wt MBP-LDB1-SSBP2 complex (or MBP-LDB1-SSBP2 mutants), 20 μM of wt GST-Pygo fragment and 20 μl dextrin sepharose high performance resin (GE Healthcare) were mixed in 100 μl of pull-down buffer. Other steps were the same as for GST pull-down assays.

### BLI assays

BLI assays were performed using the Octet RED96 system (Sartorius) to study the physical interaction between human BCL9–Pygo2, hBCL9$^{\triangle HD1}$ or hBCL9$^{\triangle HD3}$-Pygo2 complex and LDB1-SSBP2 complex containing full-length human proteins. All experiments were performed at 30 °C, and the SA biosensors were pre-equilibrated in buffer containing 20 mM Tris-HCl pH 8.0, 100 mM NaCl, 10 mM DTT, 10 mg/ml BSA for >10 min. Biotinylated BCL9-Pygo2, hBCL9$^{\triangle HD1}$ or hBCL9$^{\triangle HD3}$-Pygo2 complex were loaded onto SA biosensors (Sartorius), and SA biosensors were then dipped into a solution containing full-length LDB1-SSBP2 complex for binding measurements. The concentration gradients of LDB1-SSBP2 complex used in these BLI assays were 1000 nM, 300 nM, 100 nM and 30 nM. The interference patterns from biotinylated ACE2-immobilized biosensors with the same concentration gradients were analyzed as controls.

For the binding of different Pygo2 fragments with LDB1-SSBP2 fragments, free GST, GST-Pygo2 or GST-Pygo2 mutants were loaded onto GST biosensors (Sartorius), and GST biosensors were then quenched with free GST to block free sites on biosensors. The biosensors were dipped into wt or mutant MBP-LDB1-SSBP2 solutions for binding measurements. The concentration gradients of MBP-LDB1-SSBP2 used in these BLI assays were 0.03 μM, 0.1 μM, 0.3 μM, 1 μM, 3 μM, 10 μM, or 30 μM. The interference patterns from free GST-immobilized biosensors with the same concentration gradients were analyzed as controls. The binding affinities were determined using Octet Data Analysis 10.0 and final data analysis was done in GraphPad Prism 7.

### Cell culture and Gene editing using CRISPR/Cas9

HEK293T cells (ATCC) were cultured in DMEM with 10% v/v fetal bovine serum in a humidified incubator at 37 °C, 5 % CO₂. To generate LDB1/2 DKO or Pygo1/2 DKO cells, HEK293T cells were initially

transfected with plasmid pX458 encoding CAS9-GFP and guide RNAs targeting genomic loci of above-mentioned genes[33]. Fluorescent cells were sorted by fluorescence-activated cell sorting AriaIII (BD Biosciences), and individual clones were grown in 96-well plates. Deletions were validated by immunoblotting and sequencing (Supplementary Fig. 8).

## Western blotting

HEK293T Cells were lysed with RIPA buffer (Beyotime) supplemented with 1 mM PMSF (Sigma) and incubated for 5 min on ice, followed by 21,130 g centrifugation for 10 min. Supernatants were transferred to new tubes, protein concentrations were determined using the BCA protein assay kit (Beyotime), and 20 µg of total protein in SDS loading buffer were analyzed by SDS-PAGE. Proteins were transferred to PVDF membranes (Millipore) using the Trans-Blot Turbo Transfer system (Bio-Rad). Membranes were blocked in 5% (w/v) milk (BD) in buffer containing 50 mM Tris-HCl pH 7.4, 150 mM NaCl, 0.5% Tween-20, probed at 4 °C overnight with primary antibody: rabbit polyclonal LDB1 antibody (Abcam, AB96799, 1:1000 dilution), rabbit polyclonal Pygo2 antibody (Proteintech, 11555-1-AP, 1:600 dilution) or rabbit polyclonal histone 3 antibody (Proteintech, 17168-1-AP, 1:5000 dilution), respectively, followed by incubating for 1 h at room temperature with HRP goat anti-Rabbit IgG (H + L) (Abclonal, AS014, 1:5000 dilution). Blots were developed with Clarity Western ECL Substrate (Bio-Rad) and exposed with iBright CL1000 imaging system (Invitrogen).

## TOPflash assays

One day before transfection, HEK293T cells were inoculated in 24-well plates at a concentration of $1.5 \times 10^5$ cells/well. 100 ng of wt or mutant Pygo or LDB plasmids, 800 ng of M50 Super 8 × TOPflash plasmid (#12456, Addgene) and 100 ng of CMV-Renilla plasmid were co-transfected into cells using Lipofectamine 3000 Transfection Reagent (Thermo) according to the manufacturer's instructions. At 24 h after transfection, Wnt pathway stimulation was achieved by incubation of cells with culture medium containing 20 mM LiCl (or 20 mM NaCl as control) for 6 h unless otherwise stated. The cells were harvested for luciferase reporter assay, which was performed according to the manufacturer's protocol (Dual Luciferase Assay kit, Promega). Since HEK293T cells were transfected with M50 Super 8x TOPflash plasmid, which contains a firefly Luciferase cDNA driven by seven tandem repeats of the TCF binding site, Wnt activity was quantified by monitoring the activity of firefly Luciferase. Renilla luciferase was used as an internal control. The activity of Luciferase was detected using Spark multimode microplate reader (Tecan). Values of unstimulated wt HEK293T cells were set to 1. Raw data for TOPFlash assay readings are included in the Source Data file.

## RNA isolation, cDNA synthesis, and RT-qPCR

WT or DKO HEK293T cells were inoculated in 6-well plates at a concentration of $1 \times 10^6$ cells/well and were treated with culture medium containing 20 mM LiCl (or 20 mM NaCl as control) for 6 h. Then total RNA of these cells was isolated using TRIZOL (Thermo) according to manufacturer's protocols. Briefly, 500 µl of Trizol reagent was added directly to the culture dish to lyse the cells for each well and 50 µl of RNase-free water was added to resuspend the RNA. A total of 250 ng RNA was used to generate cDNA using PrimeScript RT Master Mix (Perfect Real Time) kit (Takara) according to the manufacturer's instructions. A 5 µl of reverse transcription reaction solution was composed of 1 µl of 5 × PrimeScript RT Master Mix (Perfect Real Time), 250 ng RNA and up to 5 µl of RNase-free water. RT-qPCR was performed using the ChamQ Universal SYBR qPCR Master Mix kit (Vazyme) with the QuantStudio 7 Real-Time PCR System (Applied Biosystems). A 20 µl of RT-qPCR reaction solution was composed of 10 µl 2 × ChamQ Universal SYBR qPCR Master Mix (Vazyme), 1 µl cDNA, 0.4 µl each of 10 µM forward and reverse qPCR primers, and 8.2 µl RNase-free water. Relative gene expression levels were estimated by the $2^{-\Delta\Delta Ct}$ method. Transcript copy numbers were normalized to the *ACTB* control gene for each sample, and values of uninduced wt HEK293T cells were set to 1. Primers are listed in Supplementary Table 2.

## RNA sequencing

RNA sequencing was performed by GENEWIZ. In brief, 1 µg of total RNA was used for library preparation, and poly(A) mRNA isolation was performed using Oligo(dT) beads. Priming was performed using Random Primers. First- and second-strand cDNA were synthesized, and the purified double-stranded cDNA was then treated to repair both ends and to add poly-dA tails in the same reaction, followed by a T-A ligation to add adaptors to both ends. Size selection of adaptor-ligated DNA was then performed using DNA Clean Beads. Each sample was then amplified by PCR, and the PCR products were sized and validated by gel electrophoresis. Libraries with different indices were multiplexed and loaded on an Illumina HiSeq instrument for sequencing using a 2 × 150 paired-end (PE) configuration according to manufacturer's instructions.

## Bioinformatics analysis

150 bp raw sequencing reads were trimmed by removing low quality sequences and adaptor sequences. Quantification of relative abundance of each transcript was reported as fragments per kilobase-of-transcript-per-million mapped reads (FPKM). GOSeq 1.34.1 was used to identify Gene Ontology (GO) terms, and KEGG (Kyoto Encyclopedia of Genes and Genomes) was used to enrich significant differential expression gene in KEGG pathways. Genes listed in Supplementary Fig. 10 were selected because of their functional relevance to Wnt signaling or because at least one item of their GO or KEGG analysis is associated with Wnt signaling, although some of them may be indirect target genes of Wnt/β-catenin. Genes with at least 1.5 fold change of expression levels as well as a P-adjusted value (q-value) <0.05 were selected as DEGs with the DESeq2 1.26.0. Heat-maps were plotted by the website (https://www.bioinformatics.com.cn), a free online platform for data analysis and visualization.

## Quantification and statistical analysis

All data are representative of at least 3 independent experiments. Unpaired two-tailed t test was used for statistical analyses. A p-value < 0.05 was considered statistically significant.

# Data availability

Atomic coordinates and structure factors of Pygo2-LDB1-SSBP2 complex generated in this study have been deposited in the Protein Data Bank (PDB) under accession code 8HIB. The crystal structure of LDB1-SSBP2 complex 6TYD was used for molecular replacement. The raw RNA-seq data generated in this study have been deposited in the NCBI Sequence Read Archive under accession PRJNA976288. Source data are provided with this paper.

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

## Acknowledgements

We appreciate assistance in data collection from the staff at the BL19U1 beamline in the National Center for Protein Science Shanghai (NCPSS). We also thank Lei Zhang at NCPSS and Yi Zhang at Shanghaitech University for technical support with the BLI assays and Zhizhi Wang for helpful discussions. This work was supported by Chinese Academy of Sciences Pilot Strategic Science and Technology Projects B grant XDB37030302, the National Laboratory of Biomacromolecules grant 2021kf01 and start-up funding from the ShanghaiTech University to W.X., the National Natural Science Foundation of China grant 32000862 to H.W., 32171218 to X.-X.Y., the China Postdoctoral Science Foundation BX20200351 to H.W., and by Cancer Research UK program grant funding (C7379/A8709, C7379/A15291 and C7379/A24639) to M.B.

## Author contributions

W.X. and X.-X.Y. designed research, H.W. performed research, H.W., M.B., X.-X.Y., and W.X. analyzed data, and H.W., M.B., X.-X.Y., and W.X. wrote the paper.

## Competing interests

The authors declare no competing interests.
