## [Peer Review File · Nature Communications]

Structural basis of the interaction between BCL9-Pygo and LDB-SSBP complexes in assembling the Wnt enhanceosomeREVIEWER COMMENTS

Reviewer #1 (Remarks to the Author):

In this structure-function study by Yang and colleagues, new structures of the beta-catenin-associated co-regulatory complex involving Pygo, BCL9 and ChiLS is presented. An interesting (and important) surface of interaction between Pygo and LDB1 and SSBP2 (which comprise the ChiLS complex) is identified. It is a tripartite mode of binding and key residues necessary for the interaction are identified through a thorough mutation analysis. Mutation of these key residues and the reconstitution of mutant protein in human cells (HEK293) were used to assess the impact of these interactions on canonical beta-catenin activity (i.e. as recruited by beta-catenin and TCF interactions).

The strengths of this study are that these structures are an exciting, important advance in understanding how Wnt signaling regulates transcription of target genes. It has been a long term goal of numerous research teams to define the structure(s) of TCF/LEF:beta-catenin complexes, a goal challenged by technical hurdles – some of which are solved here. The structures may also be an important for the ongoing effort to develop Wnt inhibitors of clinical potential. Currently, there are not many (any?) compelling, specific Wnt inhibitors that target the transcriptional regulatory complex in the nucleus. The structures presented in this manuscript could very likely be important for those efforts.

The issues with this study rest with the functional readouts for validation in human cells (Figure 6). Despite the authors' statement in the Introduction that Pygo/ChiLS interactions are required for beta-catenin:TCF activities, Figure 6A-D do not fully support that. Here are some specific comments:

1. Figure 6A-C: The TopFlash assay in panel A clearly shows that loss of Pygo or LDB impacts the ability of LiCl to activate the Wnt reporter. But cells without any Pygo still show 20X activation and cells without LDB report back 40X activation. Clearly these factors “enhance” beta-catenin:TCF/LEF activity, but they are not “required” or “essential” as stated in the Abstract and Intro. Panels B and C show some differences from the data in Panel A. Here, PygoKO cells have more than 50% of the activity of the control cell line and LDBKO cells have nearly 90% of the activity (a result that suggests LDB is not important). These modest effects on TopFlash make it difficult to conclude that the mutations that abolish binding in vitro have the expected profound loss of biological activity in vivo. Either these differences reflect natural variation in the TOPFlash assay, or the KO cell lines are somehow quite different from the cultures used for the Panel A experiment. Collectively these data do not fully support the model proposed in Figure 1 and the statements in the Introduction.

2. Figure 6D: The data shown in panel D is intriguing, but some of it is counterintuitive to the simple model that beta-catenin:BCL9:Pygo:ChiLS interactions mediating activation of Wnt target gene transcription.

a. First of all, the data analysis pipeline is not described in the legend or methods. What was the normalization method? What was the rationale behind the choice of the gene cohort used for the heatmap? Is this gene cohort comprised of Wnt pathway components selectively? Some Wnt pathway genes are Wnt targets, but others are not and some are unknown.

b. LiCl appears to activate and repress gene modules. Since LiCl inhibits GSK3beta kinase, this is plausible as GSK3b is a broadly acting kinase that affects multiple signaling pathways. In this figure, the 17 genes listed at the top are repressed with LiCl treatment – not activated. Second, PygoKO cells (Pygo^{-/-}_NaCl) have elevated expression of a 10 gene cohort and LiCl treatment reduces this expression – as if Pygo is involved in gene repression in the absence of Wnt pathway activation. Lastly, LiCl treatment has similar effects on gene expression as the PygoKO cell line except for six genes (CCND1 – TBL1XR1) and the LDBKO cells look very similar to the parental.

c. There is no doubt that Pygo and ChiLS participate with beta-catenin in activation of gene transcription; there is an extensive, well validated literature on this. But the reporter assays and the RNAseq analysis in Figure 6 do not drill down into that precisely. It would be better to analyze the data for the genes most significantly affected by loss of Pygo and ChiLS, and then devote some of the Discussion to address the positive/negative effects on gene expression. A very important discovery in

the author's study is that Pygo interacts with a dimer module of ChiLS – and therefore might very well be linked to a repressor complex like Osa/ARID1A.

3. Statistics and Bioinformatics: The statistics in Figure 6 is sparsely described. It is not clear whether the bar graphs are showing three technical replicates, or the average of three independent experiments. Panels B and C are missing statistical nomenclature to indicate what is and/or what is not statistically significant.

4. Minor comment: The model shown in Figure 1 depicts the author's concept of the overall structure of the entire activating complex. Missing in this concept is that the TCF/LEF proteins bind and bend DNA very sharply – 90°-125° - away from the protein.

Reviewer #2 (Remarks to the Author):

The Manuscript NCOMMS-22-45816 "Structural basis of the interaction between BCL9-Pygo and LDB-SSBP complexes in assembling the Wnt enhanceosome", by Hongyang Wang et.al. has clearly identified that BCL9, Pygopus (Pygo), LDB1 and SSBP form a stable core complex within the Wnt enhanceosome by biochemical assays, and defined the detailed interaction mode by crystal structure determination of a ternary complex comprising human Pygo2-NHD, LDB1 and SSBP2. They further confirmed the functional relevance of the structural findings by site-directed mutagenesis, biochemical binding assays (BLI) and double knock-out (DKO) of human cell lines lacking LDB or Pygo. Since the Wnt/ β -catenin-dependent transcriptional regulation pathway is very crucial for cancer initiation and development, the well-defined and highly conserved NPFxD motif within Pygo-NHD binding to a deep groove formed between LDB1 and SSBP2 discovered in the paper indicated to a very promising drug-binding site for design of small molecules to block Wnt/ β -catenin signaling, thus to prevent cancer development.

The overall work is well planned, the experiments successfully carried out, and the manuscript well written, but there are a few comments I would like to point out for changes.

1. In Fig. 1B, please label clearly what kind of SEC column used, better to point out the rough molecular weight (MW) with a standard curve? The correct MWs corresponding to the eluted peaks are important evidences in these experiments, the figure is not quite clear in the current setting.

2. The crystallization and structural determination between Pygo258-84 and LDB1(56-285)-SSBP2(1-94) is interesting, the ternary (Pygo258-84 and LDB1(56-285)-SSBP2(1-94)) complex and the binary LDB1-SSBP2 complex published earlier, (Wang, H. et al. Crystal structure of human LDB1 in complex with SSBP2. PNAS 117, 1042-1048 (2020)) turned out to be quite isomorphous, I wonder if the authors have tried to solve it simply by isomorphous displacement? And from the seeding procedure of crystallization one can expect the weaker occupancy of the Pygo258-84 parts, please make an electron density map of this part, and yet better to analyze the B-factors/occupancies (Q-factors) of this part.

3. In the discussion part, the author discussed that "We examined the structural features of the BCL9-Pygo-LDB1-SSBP2 complex using both negative staining EM and cryo-EM methods. It appeared that this complex per se may not adopt a tightly-packed shape (data not shown)." This part needs further elucidation and explanations, particularly in the light of crystallization experiments, is the disagreement between the EM observations and crystal structure features resulted from the different experimental techniques to resolve different interaction modes or from crystallization artifacts?

We were very happy to learn that both reviewers were excited and positive about our work. We thank both reviewers for their insightful and constructive comments and include our point-by-point response below.

Reviewer #1 (Remarks to the Author):

In this structure-function study by Yang and colleagues, new structures of the beta-catenin-associated co-regulatory complex involving Pygo, BCL9 and ChiLS is presented. An interesting (and important) surface of interaction between Pygo and LDB1 and SSBP2 (which comprise the ChiLS complex) is identified. It is a tripartite mode of binding and key residues necessary for the interaction are identified through a thorough mutation analysis. Mutation of these key residues and the reconstitution of mutant protein in human cells (HEK293) were used to assess the impact of these interactions on canonical beta-catenin activity (i.e. as recruited by beta-catenin and TCF interactions).

The strengths of this study are that these structures are an exciting, important advance in understanding how Wnt signaling regulates transcription of target genes. It has been a long term goal of numerous research teams to define the structure(s) of TCF/LEF:beta-catenin complexes, a goal challenged by technical hurdles – some of which are solved here. The structures may also be an important for the ongoing effort to develop Wnt inhibitors of clinical potential. Currently, there are not many (any?) compelling, specific Wnt inhibitors that target the transcriptional regulatory complex in the nucleus. The structures presented in this manuscript could very likely be important for those efforts.

We are very pleased with this reviewer's enthusiastic comments.

The issues with this study rest with the functional readouts for validation in human cells (Figure 6). Despite the authors' statement in the Introduction that Pygo/ChiLS interactions are required for beta-catenin:TCF activities, Figure 6A-D do not fully support that. Here are some specific comments:

1. Figure 6A-C: The TopFlash assay in panel A clearly shows that loss of Pygo or LDB impacts the ability of LiCl to activate the Wnt reporter. But cells without any Pygo still show 20X activation and cells without LDB report back 40X activation. Clearly these factors “enhance” beta-catenin:TCF/LEF activity, but they are not “required” or “essential” as stated in the Abstract and Intro.

Fair point: we have now revised our statements accordingly in the Abstract, Intro and Results, using ‘important’ or ‘support’ instead (manuscript lines 33, 48 and 240).

Panels B and C show some differences from the data in Panel A. Here, PygoKO cells have more than 50% of the activity of the control cell line and LDBKO cells have nearly 90% of the activity (a result that suggests LDB is not important). These modest effects on TopFlash make it difficult to conclude that the mutations that abolish binding in vitro have the expected profound loss of biological activity in vivo. Either these differences reflect natural variation in the TOPFlash assay, or the KO cell lines are somehow quite different from the cultures used for the Panel A experiment. Collectively these data do not fully support the model proposed in Figure 1 and the statements in the Introduction.

We thank this reviewer for pointing this out. The differences of loss of Pygo or LDB on Wnt activity between panels A and B & C in Fig. 6 are largely due to the assay time difference after adding LiCl – 4 hrs for panel A, and 6 hrs for panels B & C. Smaller differences in longer LiCl incubation time may be an effect of negative feedback of Wnt/ β -catenin signaling¹⁻³. We have now clearly marked the time difference in the figure legend.

We agree that the high levels of remaining LiCl-inducibility in the absence of Pygo (~30-50%) or LDB (~60-90%) suggests that neither enhanceosome factor is essential for these LiCl responses. We also accept that the TOPflash reporter, although convenient and readily quantifiable, is not ideal for assaying enhanceosome-dependent transcriptional activity because one of the key functions of the Wnt enhanceosome is to mediate the communication between distal enhancers and core promoter elements⁴. More than 70% of TCF4-bound regions are located at distances greater than 10 kb from the nearest annotated transcription starts⁵. In addition, the LDB scaffold protein was discovered to function in enhancer-promoter communication over long distances (i.e. tens of kilobases) in the context of cell-type specification^{6, 7}, even before LDB was implicated in Wnt responses⁴. However, the TCF binding sites in the TOPflash reporter are directly upstream of its short minimal promoter (32bp) composed of TATA box and transcription start site, separated only by a short multiple cloning site sequence (~40bp). Therefore, the reliance of TOPflash on the long-distance bridging function of LDB is at best mild (as pointed out by this reviewer) which is why we have also conducted RNA profiling of endogenous target genes (some of which might be more reliant on this function). Please note that the genes exhibiting the most pronounced reliance on LDB are the master developmental control genes^{6, 7}, but these can only be assayed in tissue of whole animals⁴. We hope that this reviewer will accept that animal-based test assays are beyond the scope of our current study whose focus is the structural determination of the core complex of the Wnt enhanceosome.

We also accept that LiCl is not an ideal stimulus of Wnt responses, given that GSK3 has many Wnt-unrelated substrates in addition to β -catenin, including a broad range of transcription factors^{8, 9} (see also below). Indeed, some of these factors may bind to distinct enhancers of Wnt target genes outside their Wnt-responsive modules, feeding into Wnt enhanceosome (which integrates signaling responses and inputs from cell lineage factors⁴). Nevertheless, LiCl has been used widely in the literature as a convenient (if somewhat inadequate) surrogate for Wnt stimulation, partly because the purification of Wnt factors with biological activity remains challenging. Combined with the readily quantifiable TOPflash assay, we think that LiCl can serve as a valid tool to mimic Wnt stimulation.

As regards Pygo, please note that several cancer cell lines with hyperactive Wnt pathway activity (e.g. colorectal, breast and lung cancer cells) show a pronounced reliance of their Wnt responses on Pygo (e.g. because some of them may be addicted to their high expression levels of Pygo), while the corresponding normal tissue cells whose Wnt pathway is less active are less reliant on Pygo¹⁰⁻¹⁴. For example, treatment of SW480 and HCT116 cells with Pygo1 RNAi causes a substantial reduction of TOPflash activity, to ~35% of controls¹⁵, and treatment of lung cancer cell lines stably transfected with Pygo2 shRNAs causes an even more significant reduction of TOPflash activity, to ~20% of controls¹⁰. Likewise, in an aggressive prostate cancer cell line, Pygo2 knockdown significantly reduced the Wnt-3A-induced TOPflash activity to ~20-60% of controls¹⁶. However, it is noteworthy that, while Pygo

knockout (or knockdown) in these cancer cell lines failed to completely abolish the transactivation of TOPflash or selected endogenous Wnt target genes, depletion of Pygo2 protein alone was sufficient to inhibit their growth and to attenuate their oncogenic progression^{11, 12, 16-19}.

We have now revised our manuscript (manuscript lines 350-366), to provide these explanations for the relatively modest reductions of the TOPflash values in the LDB and Pygo KO cells.

2. Figure 6D: The data shown in panel D is intriguing, but some of it is counterintuitive to the simple model that beta-catenin:BCL9:Pygo:ChiLS interactions mediating activation of Wnt target gene transcription.

We agree that our data shown in Fig. 6D were neither fully digested nor discussed and that they did not seem to be fully consistent with our structural and biochemical analysis. These data were obtained in HEK293T cells which, despite their epithelial features which render them suitable models for assaying Wnt responses, are of limited physiological relevance for whole animal models. However, because we did not draw any major conclusions from these data, we decided to **move Fig. 6D to the Supplement** to de-emphasize their importance, and we also **modified some of the phrasing in the corresponding manuscript sections** (see manuscript lines 267-273).

a. First of all, the data analysis pipeline is not described in the legend or methods. What was the normalization method? What was the rationale behind the choice of the gene cohort used for the heatmap? Is this gene cohort comprised of Wnt pathway components selectively? Some Wnt pathway genes are Wnt targets, but others are not and some are unknown.

We have now added the data analysis pipeline in the method section (manuscript lines 560-573). The raw gene expression data from 6 different groups (3 biological repeats for each group) underwent a Z-score normalization. The gene cohort used for the heat map is comprised of Wnt pathway components selectively. Regarding our statement that “the expression levels of 46 Wnt target genes showed in this heat map, out of 577 tested, were significantly altered in these samples (manuscript lines 266-268)”, we would like to add that all 577 genes were selected because of their functional relevance to Wnt signaling or because at least one item of their gene ontology (GO) or KEGG (Kyoto Encyclopedia of Genes and Genomes) analysis is associated with Wnt signaling, although some of them may be indirect target genes of Wnt/ β -catenin. **We now state this in the relevant sections of the text and figure legends.**

b. LiCl appears to activate and repress gene modules. Since LiCl inhibits GSK3 β kinase, this is plausible as GSK3 β is a broadly acting kinase that affects multiple signaling pathways. In this figure, the 17 genes listed at the top are repressed with LiCl treatment – not activated. Second, PygoKO cells (Pygo^{-/-}_NaCl) have elevated expression of a 10 gene cohort and LiCl treatment reduces this expression – as if Pygo is involved in gene repression in the absence of Wnt pathway activation.

Thanks for this insightful comment: we agree that the effects of LiCl treatment do not necessarily

have to be limited to Wnt signaling since GSK3 has many Wnt-unrelated substrates including a broad range of transcription factors^{8,9} some of which may feed into the Wnt enhanceosome and convert its output from positive to negative. Indeed, the Wnt enhanceosome itself can be switched from activating to repressive, i.e. some of its constituent activating components can be exchanged with transcriptional repressors to confer negative feedback regulation (see model figure in ref⁴). Furthermore, although all the genes identified by our RNA profiling (listed in our original Fig. 6D, now Supplementary Fig. 10) are Wnt-related according to GO and KEGG analysis, this list does not only include direct Wnt targets, but also indirect target genes some of which might be controlled by Wnt-induced repressors. Any of these scenarios could explain why a cohort of 10 genes in Supplementary Fig. 10 are upregulated by Pygo KO (Pygo^{-/-}_NaCl) but downregulated by LiCl treatment. Note also that similar RNA profiling analyses previously done in *Xenopus* embryos²⁰ or IMR32 cells²¹ also revealed that LiCl can upregulate or downregulate Wnt-related genes. **We now clarify this in the revised legend of Supplementary Fig. 10** (manuscript lines 905), stating that some of the genes listed in this figure may be indirect Wnt target genes.

Lastly, LiCl treatment has similar effects on gene expression as the PygoKO cell line except for six genes (CCND1 – TBL1XR1) ...

It is indeed puzzling that Pygo KO affects this small set of target genes in the same direction as LiCl treatment, but we suspect that, yet again, some of these genes may be indirect targets or subject to feedback inhibition. Detailed time courses would be required to decipher these complex regulatory circuits, experiments that we believe to be beyond the scope of our study.

... and the LDBKO cells look very similar to the parental.

As we point out at the end of our Results section, LDB DKO tends to have smaller effects on Wnt target genes than Pygo DKO, which appears inconsistent with our structural model of the Wnt enhanceosome which contains equal number of Pygo and LDB molecules (Fig. 1E), and so removal of these factors should affect target genes in the same way. We suspect that the explanation for this lies in the perdurance of stable LDB proteins, e.g. within chromatin of the HEK293T cells, following CRISPR-deletion of their genes: note that, in *Drosophila* embryos, *chip* (and *pygo*) null mutants show (similarly) severe early embryonic phenotypes but these are only visible if maternal proteins are eliminated by genetics; otherwise, maternal Chip (and Pygo) proteins perdure during numerous (>16) cell division cycles and thus sustain normal embryonic development. **We now point this out in the revised legend of this figure** (original Fig. 6D, now Supplementary Fig. 10, manuscript lines 906-914).

c. There is no doubt that Pygo and ChiLS participate with beta-catenin in activation of gene transcription; there is an extensive, well validated literature on this. But the reporter assays and the RNAseq analysis in Figure 6 do not drill down into that precisely. It would be better to analyze the data for the genes most significantly affected by loss of Pygo and ChiLS, and then devote some of the

Discussion to address the positive/negative effects on gene expression. A very important discovery in the author's study is that Pygo interacts with a dimer module of ChiLS – and therefore might very well be linked to a repressor complex like Osa/ARID1A.

Thanks for this suggestion: **we have now added a sentence in our Discussion** saying that “that a single ChiLS core complex can bind simultaneously to Pygo through one arm and to another NPF-containing protein such as ARID1 through the other: recall that ARID1 is the DNA-binding subunit of the chromatin remodeling BAF complex which switches target genes between ON and OFF states, behaving as a repressor of Wnt responses in flies and as an important tumour suppressor in humans” (manuscript lines 342-347).” To obtain evidence for such complex circuitry, an in-depth analysis of our RNA profiling would be required as well as systematic time courses combined with extensive functional validation, which we consider beyond the scope of our study. Please note that the genes whose expression changes most upon loss of Pygo or LDB are not necessarily the genes of highest physiological relevance for HEK293 cells, which is why we have listed *all* genes that are affected by Pygo DKO and LDB DKO in a statistically significant fashion (i.e. genes exhibiting >1.5 fold change in their expression) in the **Supplementary Data 1**, as we believe this could provide valuable information for future biological studies.

3. Statistics and Bioinformatics: The statistics in Figure 6 is sparsely described. It is not clear whether the bar graphs are showing three technical replicates, or the average of three independent experiments. Panels B and C are missing statistical nomenclature to indicate what is and/or what is not statistically significant.

The bar graphs show three biological replicates per experiment **as we now state in the revised figure legends** (manuscript lines 813-814). **We have also added several brackets into the figure itself**, to indicate the degree of statistical significance for various comparisons.

4. Minor comment: The model shown in Figure 1 depicts the author's concept of the overall structure of the entire activating complex. Missing in this concept is that the TCF/LEF proteins bind and bend DNA very sharply – 90°-125° - away from the protein.

Thanks for reminding us of this observation: **we have now revised Fig. 1E** to indicate that the HMG domain of TCF bends DNA, which may promote the enhancer-promoter communication of the Wnt enhanceosome, one of its key functions in gene control, as **stated in the revised figure legend** (manuscript lines 763-765).

Reviewer #2 (Remarks to the Author):

The Manuscript NCOMMS-22-45816 "Structural basis of the interaction between BCL9-Pygo and LDB-SSBP complexes in assembling the Wnt enhanceosome", by Hongyang Wang et.al. has clearly identified that BCL9, Pygopus (Pygo), LDB1 and SSBP form a stable core complex within the Wnt

enhanceosome by biochemical assays, and defined the detailed interaction mode by crystal structure determination of a ternary complex comprising human Pygo2-NHD, LDB1 and SSBP2. They further confirmed the functional relevance of the structural findings by site-directed mutagenesis, biochemical binding assays (BLI) and double knock-out (DKO) of human cell lines lacking LDB or Pygo. Since the Wnt/ β -catenin-dependent transcriptional regulation pathway is very crucial for cancer initiation and development, the well-defined and highly conserved NPFxD motif within Pygo-NHD binding to a deep groove formed between LDB1 and SSBP2 discovered in the paper indicated to a very promising drug-binding site for design of small molecules to block Wnt/ β -catenin signaling, thus to prevent cancer development.

The overall work is well planned, the experiments successfully carried out, and the manuscript well written, but there are a few comments I would like to point out for changes.

We thank this reviewer for his/her positive and constructive comments.

1. In Fig. 1B, please label clearly what kind of SEC column used, better to point out the rough molecular weight (MW) with a standard curve? The correct MWs corresponding to the eluted peaks are important evidences in these experiments, the figure is not quite clear in the current setting.

We used a Superose 6 Increase 10/300 GL column, as now stated in the revised figure legend (manuscript lines 757). The expected molecular weights (MW) of standard proteins for globally folded are shown below (Fig. 1A, from the column manufacturer's manual). Because the BCL9-Pygo-LDB1-SSBP2 complex contains extensive disordered regions, the peak positions of these complexes are expected to deviate from the standard curve (Fig. 1B; please also see our response to point #3 below). Therefore, we decided against including this standard curve in the main figure. Nonetheless, since the result in Fig. 1B (co-migration assay) and 1D (BLI assay) are highly reproducible, we consider our conclusion that BCL9, Pygo, LDB1 and SSBP can form a stable complex to be valid.

A**B**
2. The crystallization and structural determination between Pygo258-84 and LDB1(56-285)-SSBP2(1-94) is interesting, the ternary (Pygo258-84 and LDB1(56-285)-SSBP2(1-94)) complex and the binary LDB1-SSBP2 complex published earlier, (Wang, H. et al. Crystal structure of human LDB1 in complex with SSBP2. PNAS 117, 1042-1048 (2020)) turned out to be quite isomorphous, I wonder if the authors have tried to solve it simply by isomorphous displacement? And from the seeding procedure of crystallization one can expect the weaker occupancy of the Pygo258-84 parts, please make an electron density map of this part, and yet better to analyze the B-factors/occupancies (Q-factors) of this part.

Thanks for this comment: since the Pygo2-LDB1-SSBP2 ternary complex crystals were grown by microseeding, we tried isomorphous displacement, but did not succeed because of a Pygo2-induced conformational change of SSBP2 (as indicated in Supplementary Fig. S5C). As requested by this reviewer, **we have now added Supplementary Fig. S4A** with an electron density map of Pygo2(58-84) that clearly shows the density expected from the NPF motif.

As for occupancy refinement, we feel our resolution (2.45Å) is not high enough to refine occupancy and B-factor simultaneously. The Phenix manual suggests occupancy refinement for higher resolution

(usually 1.7Å or better). To illustrate the level of conformational flexibility of Pygo2, **we have now added Supplementary Fig. S4B** that shows individual atomic B factors (color-coded) for the Pygo2 peptide. This figure also shows that the NPF-motif region exhibits low B-factors, consistent with its critical anchor role for the interaction between Pygo and the LDB1-SSBP2 complex.

3. In the discussion part, the author discussed that “We examined the structural features of the BCL9-Pygo-LDB1-SSBP2 complex using both negative staining EM and cryo-EM methods. It appeared that this complex per se may not adopt a tightly-packed shape (data not shown).” This part needs further elucidation and explanations, particularly in the light of crystallization experiments, is the disagreement between the EM observations and crystal structure features resulted from the different experimental techniques to resolve different interaction modes or from crystallization artifacts?

For each protein in the BCL9-Pygo-LDB1-SSBP2 scaffold complex, available servers such as XtalPred (<https://xtalpred.godziklab.org/XtalPred-cgi/xtal.pl>) or IUPred2 (<https://iupred2a.elte.hu>) predict that the ordered portions of BCL9, Pygo2, LDB1 and SSBP2 comprise ~5%, ~21%, ~56% and ~23% of the full-length proteins, respectively. That is, ~82% of the human BCL9₂-Pygo2₂-LDB1₂-SSBP2₄ complex is predicted to be disordered and flexible. The largest known ordered structure in the BCL9-Pygo-LDB1-SSBP2 scaffold complex is the ternary (Pygo2(58-84) and LDB1(56-285)-SSBP2(1-94)) complex revealed in our current crystal structure, which totals 91.8kD – i.e. too small to be clearly visible by EM. Our cryo-EM analysis of the tetrameric complex suggests that the BCL9-Pygo-LDB1-SSBP2 complex may not adopt a globally-packed structure beyond individually folded domain structures and their constituent dual or ternary sub-complexes. Of course, we cannot rule out the possibility that additional proteins binding to the BCL9₂-Pygo2₂-LDB1₂-SSBP2₄ scaffold complex may induce additionally ordered structures in the Wnt enhanceosome. **We now state this in our revised Discussion** (manuscript lines 299-300).

Reference:

1. Logan, C.Y. & Nusse, R. The Wnt signaling pathway in development and disease. *Annual review of cell and developmental biology* **20**, 781-810 (2004).
2. Rim, E.Y., Clevers, H. & Nusse, R. The Wnt Pathway: From Signaling Mechanisms to Synthetic Modulators. *Annual review of biochemistry* **91**, 571-598 (2022).
3. Nusse, R. & Clevers, H. Wnt/ β -Catenin Signaling, Disease, and Emerging Therapeutic Modalities. *Cell* **169**, 985-999 (2017).
4. Fiedler, M. et al. An ancient Pygo-dependent Wnt enhanceosome integrated by Chip/LDB-SSDP. *eLife* **4** (2015).
5. Hatzis, P. et al. Genome-wide pattern of TCF7L2/TCF4 chromatin occupancy in colorectal cancer cells. *Molecular and cellular biology* **28**, 2732-2744 (2008).
6. Morcillo, P., Rosen, C., Baylies, M.K. & Dorsett, D. Chip, a widely expressed chromosomal protein required for segmentation and activity of a remote wing margin enhancer in *Drosophila*. *Genes Dev* **11**, 2729-2740 (1997).

7. Bronstein, R. & Segal, D. Modularity of CHIP/LDB transcription complexes regulates cell differentiation. *Fly (Austin)* **5**, 200-205 (2011).
8. Beurel, E., Grieco, S.F. & Jope, R.S. Glycogen synthase kinase-3 (GSK3): regulation, actions, and diseases. *Pharmacol Ther* **148**, 114-131 (2015).
9. Takahashi-Yanaga, F. Activator or inhibitor? GSK-3 as a new drug target. *Biochem Pharmacol* **86**, 191-199 (2013).
10. Zhou, S.Y. et al. Overexpression of Pygopus-2 is required for canonical Wnt activation in human lung cancer. *Oncol Lett* **7**, 233-238 (2014).
11. Andrews, P.G., Lake, B.B., Popadiuk, C. & Kao, K.R. Requirement of Pygopus 2 in breast cancer. *Int J Oncol* **30**, 357-363 (2007).
12. Popadiuk, C.M. et al. Antisense suppression of pygopus2 results in growth arrest of epithelial ovarian cancer. *Clin Cancer Res* **12**, 2216-2223 (2006).
13. Vafaizadeh, V. et al. The interactions of Bcl9/Bcl9L with β -catenin and Pygopus promote breast cancer growth, invasion, and metastasis. *Oncogene* **40**, 6195-6209 (2021).
14. Talla, S.B. & Brembeck, F.H. The role of Pygo2 for Wnt/ β -catenin signaling activity during intestinal tumor initiation and progression. *Oncotarget* **7**, 80612-80632 (2016).
15. Thompson, B., Townsley, F., Rosin-Arbesfeld, R., Musisi, H. & Bienz, M. A new nuclear component of the Wnt signalling pathway. *Nat Cell Biol* **4**, 367-373 (2002).
16. Lu, X. et al. An In Vivo Screen Identifies PYGO2 as a Driver for Metastatic Prostate Cancer. *Cancer Res* **78**, 3823-3833 (2018).
17. Zhang, D. et al. Pygo2 as a novel biomarker in gastric cancer for monitoring drug resistance by upregulating MDR1. *J Cancer* **12**, 2952-2959 (2021).
18. Liu, R. et al. Pygopus 2 promotes kidney cancer OS-RC-2 cells proliferation and invasion in vitro and in vivo. *Asian J Urol* **2**, 151-157 (2015).
19. Liu, Y. et al. Abnormal expression of Pygopus 2 correlates with a malignant phenotype in human lung cancer. *BMC Cancer* **13**, 346 (2013).
20. Ding, Y. et al. Spemann organizer transcriptome induction by early beta-catenin, Wnt, Nodal, and Siamois signals in *Xenopus laevis*. *Proc Natl Acad Sci U S A* **114**, E3081-e3090 (2017).
21. Duffy, D.J., Krstic, A., Schwarzl, T., Higgins, D.G. & Kolch, W. GSK3 inhibitors regulate MYCN mRNA levels and reduce neuroblastoma cell viability through multiple mechanisms, including p53 and Wnt signaling. *Mol Cancer Ther* **13**, 454-467 (2014).

REVIEWERS' COMMENTS

Reviewer #1 (Remarks to the Author):

This revision by Yang and colleagues, provides new data, text modifications, new text on statistical and bioinformatic pipelines, and thoughtful responses to this reviewer's concerns. The specifics are listed below, but overall, this reviewer agrees with the author's perspectives and agrees that this is a complete study with important new discoveries in the mechanisms by which Wnt signaling regulates target gene expression. There are a few specifics below for purposes of completing the circle:

1. The strengths of this study remain with the newly discovered structures. They truly represent an advance in our understanding of how Wnt signaling regulates transcription of target genes. In my view, the discovery through the structural analysis that activators and repressors could hypothetically be co-recruited, or at the very least undergo dynamic exchange presents numerous opportunities for probing the complexities of gene regulation. These structures are surely to trigger the next phase of research in multiple laboratories.

2. It also remains, that the structures will be helpful in thinking about and designing small molecular disruptors for potential clinical use.

3. The limitation of the TOPFLASH assay, particularly for studying the role of factors that enable enhancer looping and other long-range modes of regulation, is absolutely correct – I agree with the point that the authors are making. LDB actions will be masked, or muted in this type of assay. I also agree with the authors in that probing LDB function in the context of long-range activation represents the next research phase and this lies beyond the study presented here. It would be an entire project (s) on its own. What was more important was to make sure to acknowledge the limited effects of LDB in the assays presented, and to discuss possible explanations including the limitation of the transient reporter assay so readers would be sure to have that perspective.

4. The unexpected, counterintuitive results with endogenous gene expression also presents a new question to address for the next phase of research (and it is appropriate to relegate the results to a supplemental figure). Feedback regulation and indirect effects all come in to play when endogenous genes are assessed – particularly at time frames of hours/days used in this study. Therefore these questions represent important and intriguing new questions for the future, making this study likely to be highly cited.

We were very happy to learn that reviewer #1 agrees with our perspectives and agrees that this is a complete study with important new discoveries in the mechanisms by which Wnt signaling regulates target gene expression. We include our point-by-point response below.

Reviewer #1 (Remarks to the Author):

This revision by Yang and colleagues, provides new data, text modifications, new text on statistical and bioinformatic pipelines, and thoughtful responses to this reviewer's concerns. The specifics are listed below, but overall, this reviewer agrees with the author's perspectives and agrees that this is a complete study with important new discoveries in the mechanisms by which Wnt signaling regulates target gene expression. There are a few specifics below for purposes of completing the circle:

We are very pleased with this reviewer's enthusiastic comments.

1. The strengths of this study remain with the newly discovered structures. They truly represent an advance in our understanding of how Wnt signaling regulates transcription of target genes. In my view, the discovery through the structural analysis that activators and repressors could hypothetically be co-recruited, or at the very least undergo dynamic exchange presents numerous opportunities for probing the complexities of gene regulation. These structures are surely to trigger the next phase of research in multiple laboratories.

We agree and thank this reviewer for their positive and constructive comments.

2. It also remains, that the structures will be helpful in thinking about and designing small molecular disruptors for potential clinical use.

We agree and thank this reviewer for this positive comment.

3. The limitation of the TOPFLASH assay, particularly for studying the role of factors that enable enhancer looping and other long-range modes of regulation, is absolutely correct – I agree with the point that the authors are making. LDB actions will be masked, or muted in this type of assay. I also agree with the authors in that probing LDB function in the context of long-range activation represents the next research phase and this lies beyond the study presented here. It would be an entire project (s) on its own. What was more important was to make sure to acknowledge the limited effects of LDB in the assays presented, and to discuss possible explanations including the limitation of the transient reporter assay so readers would be sure to have that perspective.

We are very happy that this reviewer agrees with our point regarding the limitation of

the TOPflash assay for studying the role of LDB. Note also that we have added an explanation regarding the limited effects of LDB in the TOPflash assay (see manuscript lines 323-335).

4. The unexpected, counterintuitive results with endogenous gene expression also presents a new question to address for the next phase of research (and it is appropriate to relegate the results to a supplemental figure). Feedback regulation and indirect effects all come in to play when endogenous genes are assessed – particularly at time frames of hours/days used in this study. Therefore these questions represent important and intriguing new questions for the future, making this study likely to be highly cited.

We thank this reviewer for suggesting that our study is likely to be highly cited.